# SLM: End-to-end Feature Selection via Sparse Learnable Masks

## Abstract

Feature selection has been widely used to alleviate compute requirements during training, elucidate model interpretability, and improve model generalizability. We propose SLM – Sparse Learnable Masks – a canonical approach for end-to-end feature selection that scales well with respect to both the feature dimension and the number of samples. At the heart of SLM lies a simple but effective learnable sparse mask, which learns which features to select, and gives rise to a novel objective that provably maximizes the mutual information (MI) between the selected features and the labels, which can be derived from a quadratic relaxation of mutual information from first principles. In addition, we derive a scaling mechanism that allows SLM to precisely control the number of features selected, through a novel use of sparsemax. This allows for more effective learning as demonstrated in ablation studies. Empirically, SLM achieves state-of-the-art results against a variety of competitive baselines on eight benchmark datasets, often by a significant margin, especially on those with real-world challenges such as class imbalance.

## 1 Introduction

In many machine learning scenarios, a significant portion of the input features may be irrelevant to the output, especially with modern data management tools allowing easy construction of large-scale datasets by joining features from many different data sources. "Feature selection", or filtering the most relevant features for the downstream task, is an everlasting problem, with many methods proposed to date and used (Guyon & Elisseeff, 2003; Li et al., 2017; Dash & Liu, 1997).

Feature selection can bring a multitude of benefits. Smaller number of features can yield superior generalization and hence better test accuracy, by minimizing reliance on spurious patterns that do not hold consistently (Sagawa et al., 2020), and not wasting model capacity on less relevant features. In addition, reducing the number of input features can decrease the computational complexity and cost for deployed models, as the models need to learn from smaller dimensional input data, and hence require reduced infrastructure. Lastly, feature selection facilitates interpretability, as it sheds light on which features are most relevant for the downstream task.

Given the wide applicability of feature selection, how can one select the target number of features in an efficient, effective way? §2 summarizes numerous approaches. For superior task accuracy, one desired property is that the feature selection method should consider the predictive model itself, as the optimal set of features would depend on the mapping between the input data and output labels. Such end-to-end learning methods have been approached in different ways, such as via sparse regularization and its extensions (Lemhadri et al., 2019), concrete autoencoders (Abid et al., 2019), or learned stochastic gates (Yamada et al., 2020), among others. These constitute different ways to tackle the fundamental challenge, making the feature selection operation differentiable.

In this work, we present SLM – **S**parse **L**earnable **M**asks – a novel soft approximator for the feature selection with end-to-end learning. SLM is designed to be scalable and easily-adaptable into a variety of models, and yields superior task performance. At its heart, SLM learns a sparse mask to filter out non-selected features. This mask gives rise to a novel mutual information (MI) objective, which provably maximizes the MI between

the labels and the selected features, based on a novel quadratic relaxation of the MI (§4). Furthermore, SLM proposes a scaling mechanism for sparsemax (Martins & Astudillo, 2016) to precisely control the number of features selected, which when allowed to vary during training, enables more effective learning as demonstrated in ablation studies. SLM scales well with respect to both the feature dimension and the number of features. Specifically, as detailed in §4.4, it scales $O(n)$ with respect to the dataset size $n$, and $O(F \log F)$ with respect to the feature dimension $F$. SLM can be integrated into any deep learning architecture, given the optimization is gradient-descent based for joint training. We demonstrate state-of-the-art task performance with SLM, against a myriad of competitive baseline methods, on nine datasets from wide-ranging domains.

## 2 Related work

**Feature selection methods**: Numerous methods have been studied for feature selection, and broadly fall under three categories (Guyon & Elisseeff, 2003):

- **Wrappers** recompute the predictive model for each subset of features. As exhaustive search is NP-hard and computationally intractable, efficient search strategies such as forward selection or backward elimination have been developed. For instance, HSIC-Lasso (Yamada et al., 2014) proposes a feature-wise kernelized Lasso for capturing non-linear dependencies. *Wrappers* are difficult to integrate with modern deep learning, as the training complexity gets prohibitively large.
- **Filters** select subsets of variables as a pre-processing step, independent of the predictive model. (Gu et al., 2012) developed the Fisher score, which selects features to maximize (minimize) the distances between data points in different (same) classes in the space spanned by the selected features. Principal feature analysis (PFA) (Lu et al., 2007b) selects features based on principal component analysis. (Pan et al., 2020) uses adversarial validation to select the features, based on how much their characteristics differ between training and test splits, as a way to improve robustness. There are also various methods based on MI maximization (Ding & Peng, 2005), selecting features independent of the predictive model (unlike SLM). CMIM (Fleuret, 2004) maximizes the *conditional* MI between selected features and the class labels to account for feature inter-dependence. On the other hand, JMIM (Bennasar et al., 2015) maximizes the *joint* MI between class labels and the selected features, while addressing overconfidence in features that correlate with already-selected features, with greedy search that selects features one at a time. (Zadeh et al., 2017) formulates feature selection as a diversity maximization problem using a MI-based metric amongst features. The fundamental disadvantage of filter-based methods, of not being optimized with the predictive models, results in them often yielding suboptimal performance.
- **Embedded** methods combine selection into training and are usually specific to given predictive models. Lasso regularization (Tibshirani, 1996) employs feature selection by varying the strength of the L1 regularization. (Feng & Simon, 2017) extends this idea by proposing an input-sparse neural network, where the input weights are penalized using the group Lasso penalty. (Lemhadri et al., 2019) selects only a subset of the features using input-to-output residual connections, allowing features to have non-zero weights only if their skip-layer connections are active. Concrete Autoencoder (Abid et al., 2019) proposes an unsupervised feature selector based on using a concrete selector layer as the encoder and using a standard neural network as the decoder. FsNet (Singh et al., 2020) uses a concrete random variable for discrete feature selection in a selector layer and a supervised deep neural network regularized with the reconstruction loss. STG (Yamada et al., 2020) learns stochastic gates with a probabilistic relaxation of the count of the number of selected features, it selects features and learns task prediction end-to-end.

**Masking in deep neural networks**: Masking the input to control information propagation is a commonly-used approach in deep learning. Attention-based architectures, such as Transformer (Vaswani et al., 2017) and Perceiver (Jaegle et al., 2021), show strong results across many domains, with learnable key and query representations, whose alignment yields the masks that control the contribution of corresponding value representations. While these effectively reweigh the input, they typically do not completely mask out (i.e. yielding zero attention weight) any part of the input. Towards this end, various works have focused on bringing sparsity into masking, such as based on thresholding (Zhao et al., 2019) or sparse normalization (Correia et al., 2019). TabNet (Arik & Pfister, 2019) directly generates sparse attention masks and applies them sequentially to input data, which can perform sample-dependent feature selection. (Correia et al., 2020) achieves sparsity in latent distributions in neural networks, by using sparsemax and its structured analogs,

allowing for efficient latent variable marginalization. (Lei et al., 2016) and (Bastings et al., 2019) learn Bernoulli variables, which are analogous to SLM's feature mask but in a local setting, for extractive rationale prediction in text. (Paranjape et al., 2020) extends these ideas by proposing to control sparsity by optimizing the Kullback–Leibler (KL) divergence between the mask distribution and a prior distribution with controllable sparsity levels. (Guerreiro & Martins, 2021) develops a flexible mask-based rationale extraction mechanism using a constrained structured prediction algorithm on factor graphs. All these perform *sample-wise*, not global, input selection. In this work, our goal is to explore *global* feature selection. When training and testing datasets perfectly align in distribution, local feature selection can give superior performance due to its input-dependence. However, there is rarely such perfect alignment, and global selection provides robustness benefits when there is distribution shift between training and test datasets, in addition to allowing more computational efficiency by globally removing features.

## 3 Methods

Algorithm 1 describes SLM's end-to-end feature selection and task learning. The predictor $f_\theta$ can be any gradient-descent based model, such as an MLP, with a task-specific loss function $l$ such as the cross entropy for classification or mean absolute error (MAE) for regression. The following sections present SLM's key components in detail.

**Notation.** Throughout this work, we let $\mathbf{X} \in \mathbb{R}^{n \times d}$ denote the input data, $\mathbf{X_{sp}} \in \mathbb{R}^{n \times d}$ the selected features, and $\boldsymbol{m}_{\mathrm{sp}} \in \mathbb{R}^d$ the learned sparse feature selection mask. We use $\odot$ to denote element-wise multiplication between each input sample and $\boldsymbol{m}_{\mathrm{sp}}$: $\mathbf{X_{sp}} = \mathbf{X} \odot \boldsymbol{m}_{\mathrm{sp}}$. We let $F_t$ denote the number of selected features at step $t$, and $N$ the total training steps. Furthermore, $I(X, Y)$ denotes the mutual information between $X$ and $Y$, and $I_q(X, Y)$ is its quadratic relaxation. $f_\theta$ denotes the task predictor used on the selected features.

**Overview of SLM-based feature selection.** As outlined in Algorithm 1, SLM first learns a non-sparse mask $\boldsymbol{m} \in \mathbb{R}^d$, which is turned into a sparse vector by applying the sparsemax operator (Martins & Astudillo, 2016), described in §3.1. We present a novel application of sparsemax that provably achieves output sparsity at desired level exactly, for which we propose dynamically computing a scaling constant for the mask $\boldsymbol{m}$, detailed in §3.2. SLM uses the resulting sparse mask to zero out non-selected features. The mask sparsity gradually increases throughout training to facilitate model convergence (§3.3). Finally, a predictor model $f_\theta$ on the selected features is trained using the dataset task loss and a novel mutual information (MI) loss, which is derived from first principles in §4. The following sections explain the important constitutents of SLM in detail.

---

**Algorithm 1** Training for SLM-based feature selection.

---

**Input:** Input data $\mathbf{X}$ with target labels $\mathbf{Y}$
**Input:** Total training steps $N$
**Initialize:** Learnable mask argument $\boldsymbol{m} \leftarrow$ all ones vector
**for** $t = 1$ **to** $N$ **do**
    Obtain the number of selected features $F_t$ using Eq 4 in §3.3 for step $t$.
    Compute scaling parameter $m$ for mask argument to control exact mask sparsity, using Lemma 3.2 in §3.2.
    Generate **sparse mask** $\boldsymbol{m}_{\mathrm{sp}} = \mathrm{sparsemax}(m * \boldsymbol{m})$.
    **Select** and **weight** input features with mask: $\mathbf{X_{sp}} = \mathbf{X} \odot \boldsymbol{m}_{\mathrm{sp}}$. Non-selected features are zeroed out.
    Input the selected features into the predictor $f_\theta(\mathbf{X_{sp}})$ for the downstream task.
    Compute **dataset task loss** $l(\mathbf{X}_{sp}, \mathbf{Y})$ and **MI loss** $E(\mathbf{X}_{sp}, \mathbf{Y})$ in Eq 9 from §4.3.
    Use the combined loss to **update** the model parameters $\theta$ and $\boldsymbol{m}$.

---

### 3.1 Mask sparsity via projection onto probability simplex

SLM selects features by learning a mask $\boldsymbol{m}_{\mathrm{sp}} \in \mathbb{R}^d$, and zeroing out the features in the input $\boldsymbol{X} \in \mathbb{R}^{n \times d}$ whose corresponding mask entries are zero. We use sparsemax normalization (Martins & Astudillo, 2016) to achieve sparsity in $\boldsymbol{m}$. Sparsemax achieves sparsity in its output by returning the Euclidean projection of the

input vector $\boldsymbol{v} \in \mathbb{R}^d$ onto the probability simplex $\Delta^{d-1} := \{f \in \mathbb{R}_{\geq 0}^d \mid \sum_k f_k = 1\}$:

$$\mathrm{sparsemax}(\boldsymbol{v}) := \mathrm{argmin}_{\boldsymbol{p} \in \Delta^{d-1}} \|\boldsymbol{p} - \boldsymbol{v}\|^2. \tag{1}$$

We apply sparsemax to the mask argument $\boldsymbol{m} \in \mathbb{R}^d$ to obtain sparse feature mask:

$$\boldsymbol{m}_{\mathrm{sp}} := \mathrm{sparsemax}(\boldsymbol{m}). \tag{2}$$

In particular $\boldsymbol{m}_{\mathrm{sp}} \in \mathbb{R}_{\geq 0}^d$. Compared to approaches like softmax normalization employed with thresholding, the probability simplex projection in sparsemax$(\boldsymbol{v})$ scales the top values in $\boldsymbol{v}$ so they are more equidistributed over $[0, 1]$. This equidistribution leads to greater feature weight separation, encouraging the model to discriminate amongst the features. Additional discussion on the properties of SLM sparsemax can be found in §A.5.

### 3.2 Mask scaling to yield desired number of selected features

Following its formulation, sparsemax does not yield a predetermined number of non-zero elements, as the sparsity depends on the location on the probability simplex $\Delta^{d-1}$ that $\boldsymbol{v}$ projects onto. For a non-uniform vector $\boldsymbol{v} \in \mathbb{R}^d$, we can adjust its projection onto $\Delta^{d-1}$ by multiplying $\boldsymbol{v}$ by a positive scalar. In particular, a sufficiently large scalar increases the sparsity, while a sufficiently small scalar decreases the sparsity. To illustrate this, we provide a simple example in Fig 1.

*Example* 3.1 (Adjusting sparsemax$(\boldsymbol{v})$ sparsity by scaling). The probability simplex $\Delta^1$ in $\mathbb{R}^2$ is the line connecting $(0, 1)$ and $(1, 0)$, with these two points as the simplex boundary. Let $\boldsymbol{v} = (x, y)$ be a point in $\mathbb{R}^2$, and $(z, w)$ its projection onto $\Delta^1$. We show that by varying multiplier $m$, sparsemax$(m\boldsymbol{v})$ would have a varying degree of sparsity. The projection $(z, w) = \mathrm{sparsemax}((x, y))$ is the unique point that satisfies $(z, w) = \mathrm{argmin}_{(z,w)}(\|y - w\|^2 + \|x - z\|^2)$, $(z, w)$ element-wise positive, and $z + w = 1$. As we scale $(x, y)$ with $m$, sparsemax$(m(x, y)) = argmin_{(z,w)}(\|my - w\|^2 + \|mx - z\|^2)$. This projection distance expands to

$$d(z, w) := \|my - w\|^2 + \|mx - z\|^2$$
$$= m^2 y^2 - 2myw + w^2 + m^2 x^2 - 2mxz + z^2$$

Hence, $d(0, 1) - d(0.5, 0.5) = mx - my + 0.5$ (where $(0.5, 0.5)$ is the midpoint of the simplex), which means that for any $(x, y)$ and $m$ with $y > x$, sparsemax$(m(x, y))$ is closer to $(0, 1) \in \Delta^1$ whenever $m > 1/(2(y - x))$, and closer to $(0.5, 0.5)$ otherwise. Since projection is linear, this means varying the multiplier $m$ varies the sparsity of sparsemax$((x, y))$. Figure 1 illustrates a concrete instance of scaling in the 2D case.

This example conveys the intuition that larger multipliers lead to sparser outputs. More generally, one can show:

**Lemma 3.2.** *Given a non-uniform vector $\boldsymbol{v} \in \mathbb{R}^d$, to obtain $F$ nonzero elements in sparsemax($\boldsymbol{v}$), $\boldsymbol{v}$ should be multiplied with the scalar*

$$m = \begin{cases} \left( \sum_{i=1}^{F+1} v_{(i)} - (F+1) \cdot v_{(F+1)} \right)^{-1} & \textit{if } |sparsemax(\boldsymbol{v}) > 0| > F \\ \left( \sum_{i=1}^{F} v_{(i)} - F \cdot v_{(F)} \right)^{-1} & \textit{if } |sparsemax(\boldsymbol{v}) > 0| < F, \end{cases} \tag{3}$$

*where $v_{(1)} \geq v_{(2)} \ldots \geq v_{(d)}$ denote sorted elements of $\boldsymbol{v}$ in descending order.*

The proof can be found in §A.3. Lemma 3.2 allows us to scale the mask to achieve the desired number of non-zero features. Note that since sparsemax has a particular Fenchel-Young loss (Blondel et al., 2020), scaling its argument by $m$ is equivalent to scaling the regularizer by $1/m$ in the Fenchel-Young formulation (Blondel et al., 2020; Peters et al., 2019).

### 3.3 Tempering feature sparsity to facilitate convergence

Starting training on only a randomly selected subset of features likely leads to suboptimal learning in the initial steps, and if feature selection converges before the predictor converges, the predictor would be trained

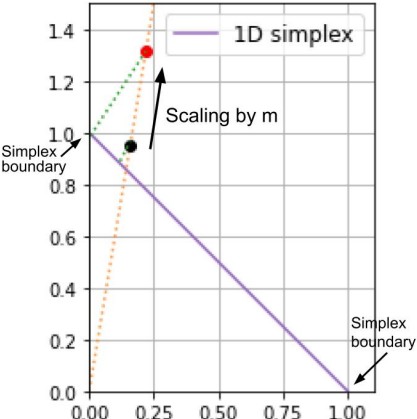

Figure 1: Mask scaling for sparsemax: We show an illustrative example on how varying the multiplier varies the sparsity. Scaling $\boldsymbol{v}$ from the *black* to the *red* point moves its projection (green dotted line) onto $\Delta^1$ closer to the simplex boundary, increasing sparsemax($\boldsymbol{v}$) sparsity, as the $x$ coordinate of the projection becomes 0. The orange dotted line indicates the path of scaling by a constant. Example 3.1 contains concrete calculations demonstrating this phenomenon.

with suboptimal features. To alleviate these and improve training stability, we propose gradually decreasing the number of features selected until reaching the target $F_N$:

$$F_t = \begin{cases} F_0 - t/N_{tmp}(F_0 - F_N) & \text{if } t < N_{tmp} \\ F_N & \text{if } t \geq N_{tmp}, \end{cases} \tag{4}$$

where $F_t$ is the number of selected features at step $t$, $N_{tmp}$ is the tempering threshold. In our experiments, we simply set $N_{tmp} = N/2$ as it's observed to be a reasonable value across a wide range of datasets (as before $N$ denotes the total number of training steps). To further stabilize training, instead of continuously decreasing the number of features, we decrease the number of features at five evenly spaced steps. This tempering allows the model to learn from more than the final target number of features during training – an advantage not shared by baseline methods. Furthermore, learning from all features initially likely provides a more robust initialization compared to starting learning with the target number of features, as the randomness in the initial selection is seldom optimal.

## 4 Mutual information maximization

As an inductive bias to the model that accounts for sample labels during feature selection, we propose to maximize the mutual information (MI) between the distribution of the selected features and the distribution of the labels. Specifically, we *condition* the MI on the probability that a feature is selected, as given by the mask $\boldsymbol{m}$. This stands in contrast to prior MI-based feature selection works such as (Fleuret, 2004; Bennasar et al., 2015), which yield binary decisions on whether to select a feature.

Let $X$ **denote** the random variable representing the features, and $Y$ the random variable representing the labels, with value spaces $X \in \mathcal{X}$ and $Y \in \mathcal{Y}$. We let $X$ and $Y$ be discrete, following a long line of research on mutual information and entropy estimation that focuses on the case where the random variables live in the discrete space (Paninski, 2003; Kraskov et al., 2004; Valiant & Valiant, 2011; Han et al., 2015; Jiao et al., 2015; Wu & Yang, 2016), this is because 1) many variables in machine learning are indeed discrete, e.g. vocabulary index in NLP, categorical variables such as nationality, gender, etc, and 2) MI estimation in the continuous case can be reduced to the discrete case via binning and taking a limit (Paninski, 2003; Kraskov et al., 2004).

Feature selection methods based on maximizing either the conditional or the joint MI between selected features and labels require the computation of an exponential number of probabilities, the optimization of which is intractable (Fleuret, 2004). Therefore, we propose an end-to-end differentiable, quadratic relaxation

for MI. When we model $X$ and $Y$ as random variables, their MI $I(X,Y)$ can be defined and reformulated as:

$$I(X,Y) := \sum_{x \in \mathcal{X}} \sum_{y \in \mathcal{Y}} P_{X,Y}(x,y) \log \frac{P_{X,Y}(x,y)}{P_X(x)P_Y(y)}$$

$$= \left( \sum_{x \in \mathcal{X}} \sum_{y \in \mathcal{Y}} P_{X,Y}(x,y) \log \frac{P_{X,Y}(x,y)}{P_X(x)} \right) - \sum_{y \in \mathcal{Y}} P_Y(y) \log P_Y(y), \quad (5)$$

where the second step derives from marginalizing over $\mathcal{X}$. Since the second term above does not depend on features $X$, it can be ignored during optimization.

### 4.1 Quadratic relaxation

We propose a quadratic relaxation $I_q(X,Y)$ of Eq 5 to simplify $I(X,Y)$ and its optimization, while retaining much of its properties:

$$I_q(X,Y) := \left( \sum_{x \in \mathcal{X}} \sum_{y \in \mathcal{Y}} P_{X,Y}(x,y)^2 / P_X(x) \right) - \sum_{y \in \mathcal{Y}} P_Y(y)^2. \quad (6)$$

Here, terms of the form $p \log q$ are relaxed to $pq$. Note that both $p \log q$ and $pq$ are convex with respect to $p$ and $q$, and hence have the same correlation behavior with respect to $p$ and $q$. From an optimization perspective, $I_q(X,Y)$ is a good approximation of $I(X,Y)$ where $P_{X,Y}(X,Y)/P_X(x)$ and $P_Y(y)$ in Eq 6 lie in the neighborhood $(1-\delta, 1+\delta)$. In this neighborhood, using Taylor expansion: $\log(q) = \log(q_0) + (q-q_0)/q_0 - (q-q_0)^2/2q_0^2 + \cdots$ When $q_0 = 1$, this becomes $\log(q) \approx (q-1) - (q-1)^2/2 = -3/2 + 2q - q^2/2$, hence, $p \log(q)$ has the second order approximation $-3p/2 + 2pq$ (or $-3p/2 + 2p^2$ when $p=q$). Applying this to Eq 5, $p$ is $P_{X,Y}(x,y)$ in the first term and $P_Y(y)$ in the second. Since both $P_{X,Y}(x,y)$ and $P_Y(y)$ are probabilities, and hence must sum to 1 across the label space for any given sample, the linear term $-3p/2$ does not affect gradient descent optimization. Normalization is a hard constraint enforced during training that supersedes this linear term in the objective. Therefore, during optimization, $P_{X,Y}(x,y) \log(P_{X,Y}(x,y)/P_X(x))$ and $P_{X,Y}(x,y)^2/P_X(x)$, and thus $I_q(X,Y)$ and $I(X,Y)$, agree on their second order approximation. Note that the proposed relaxation is a variant of the commonly-used quadratic approximation based on Taylor's theorem (Shafer, 1974; Hsieh et al., 2011).

### 4.2 Relating MI $I_q(X,Y)$ to model error $E(X,Y)$

Next we connect $I_q(X,Y)$ with the model's predictions using Lagrange multipliers. Let $R(x,y) : \mathcal{X} \times \mathcal{Y} \to [0,1]$ denote the model's probability output for sample $x$ and outcome $y$. Below, we model the discrete label case, e.g. for classification; the case where labels are continuous can be done by first discretizing the continuous label space (Fleuret, 2004), and then taking the limit as the discretization becomes infinitesimal. §A.4 contains further details. First, we define the quadratic error term $E(X,Y)$ in terms of $R(x,y)$, and expand:

$$E(X,Y) := \sum_{x \in \mathcal{X}, y \in \mathcal{Y}} P_{X,Y}(x,y) \left( (1 - R(x,y))^2 + \sum_{y' \in \mathcal{Y} \setminus y} R(x,y')^2 \right)$$

$$= \sum_{x \in \mathcal{X}, y \in \mathcal{Y}} P_{X,Y}(x,y) \left( 1 - 2R(x,y) + R(x,y)^2 + \sum_{y' \in \mathcal{Y} \setminus y} R(x,y')^2 \right)$$

$$= \sum_{x \in \mathcal{X}, y \in \mathcal{Y}} P_{X,Y}(x,y) - 2 \sum_{x \in \mathcal{X}, y \in \mathcal{Y}} P_{X,Y}(x,y)R(x,y)$$

$$+ \sum_{x \in \mathcal{X}, y \in \mathcal{Y}, y' \in \mathcal{Y}} P_{X,Y}(x,y)R(x,y')^2 \quad \triangleleft \text{Combine last two terms and expand.}$$

$$= 1 - 2 \sum_{x \in \mathcal{X}, y \in \mathcal{Y}} P_{X,Y}(x,y)R(x,y) + \sum_{x \in \mathcal{X}, y' \in \mathcal{Y}} P_X(x)R(x,y')^2 \quad \triangleleft \text{Marginalize.} \quad (7)$$

**Theorem 4.1.** *Let $X$ and $Y$ denote the random variables representing the features and labels, respectively, and $\mathcal{Y}$ the value space for $Y$, then minimizing the optimum error $E(X,Y)$ in the model space $\{f : X \to Y\}$ is equivalent to maximizing the quadratic relaxation of mutual information $I_q(X,Y)$. More specifically,*

$$\min_{f: \mathcal{X} \to \mathcal{Y}} E(X,Y) = 1 - \sum_{y \in \mathcal{Y}} P_Y(y)^2 - I_q(X,Y).$$

The proof utilizes Lagrange multipliers to solve for the optimal model predictions in terms of $P_{X,Y}(x,y)$ and $P_X(x)$, this can then be used to express the optimum objective $E(X,Y)$ as a function of $I_q(X,Y)$. The full proof can be found in §A.4.

### 4.3 Application to feature selection

Now, we apply this finding concretely to feature selection, by selecting a given number of features that minimize $E(X,Y)$. Given a dataset, let $\mathcal{I}$ denote the index set of the dataset samples, $\mathcal{J}$ the index set of the features, and $\mathcal{L}$ the set of possible labels. Let $\mathcal{S} \subset \mathcal{J}$ denote the **index set** of features selected, $X_i^{\mathcal{S}}$ the random variable representing a selected subset of features for the $i^{th}$ sample, and $Y_i$ the random variable representing the label for the $i^{th}$ sample. Then, the joint probability can be written as $P_{X,Y}(x,y) = |\{i \in \mathcal{I}|X_i^{\mathcal{S}} = x, Y_i = y\}|/|I|$. Plugging this into the definition of $E(X,Y)$ we obtain:

$$
\begin{aligned}
E(X,Y) &:= \sum_{x \in \mathcal{X}, y \in \mathcal{Y}} P_{X,Y}(x,y) \left( (1 - R(x,y))^2 + \sum_{y' \neq Y_i} R(x,y')^2 \right) \\
&= \sum_{x \in \mathcal{X}, y \in \mathcal{Y}} \frac{|\{i \in \mathcal{I} \mid X_i^{\mathcal{S}} = x, Y_i = y\}|}{|\mathcal{I}|} \left( (1 - R(X_i^{\mathcal{S}}, Y_i))^2 + \sum_{y \neq Y_i} R(X_i^{\mathcal{S}}, y)^2 \right) \\
&= \sum_{i \in \mathcal{I}} \left( (1 - R(X_i^{\mathcal{S}}, Y_i))^2 + \sum_{y \neq Y_i} R(X_i^{\mathcal{S}}, y)^2 / |\mathcal{I}| \right)
\end{aligned}
\tag{8}
$$

During training, Eq. 8 is minimized under the following consistency constraint: for two samples $i_1$ and $i_2$ that have the same values in the selected features, i.e. $X_{i_1}^{\mathcal{S}} = X_{i_2}^{\mathcal{S}}$, their model predictions must be the same, i.e. $R(X_{i_1}^{\mathcal{S}}, Y_{i_1}) = R(X_{i_2}^{\mathcal{S}}, Y_{i_2})$. To encourage the model to satisfy this constraint, we turn it into a soft consistency regularization term $r_{cs}$, converting constrained optimization to unconstrained optimization with regularization:

$$
r_{\text{cs}} := \sum_{\{i_1, i_2\} \in \mathcal{I}^2, i_1 < i_2} P(X_{i_1}^{\mathcal{S}} = X_{i_2}^{\mathcal{S}}) \left( R(X_{i_1}^{\mathcal{S}}, Y_{i_1}) - R(X_{i_2}^{\mathcal{S}}, Y_{i_2}) \right)^2,
$$

where $P(X_{i_1}^{\mathcal{S}} = X_{i_2}^{\mathcal{S}})$ is the probability that the samples $X_{i_1}$ and $X_{i_2}$ take the same values in the selected feature set $\mathcal{S}$.

Let the learned mask consists of probabilities $\boldsymbol{m} = \{p_j\}_{j \in \mathcal{J}}$, i.e. $p_j$ is the probability that feature $j$ is selected, then $P(X_{i_1}^{\mathcal{S}} = X_{i_2}^{\mathcal{S}}) = \prod_{X_{i_1}^{(j)} \neq X_{i_2}^{(j)}} (1 - p_j)$, i.e. $P(X_{i_1}^{\mathcal{S}} = X_{i_2}^{\mathcal{S}})$ is the product over probabilities that feature $j$ is not selected, if $X_{i_1}$ and $X_{i_2}$ differ at feature $j$. (The difference in a feature that is not selected does not contribute to $P(X_{i_1}^{\mathcal{S}} = X_{i_2}^{\mathcal{S}})$). In this probabilistic form, the consistency regularizer also encourages the selection of features with diverse ranges, since it encourages high $p_j$ for the features with many $X_{i_1}^{(j)} \neq X_{i_2}^{(j)}$ pairs. Therefore, the regularized objective to maximize the MI $I(X,Y)$ between the selected features and the labels becomes:

$$
E(X,Y) = \sum_{i \in \mathcal{I}} \left( (1 - R(X_i^{\mathcal{S}}, Y_i))^2 + \sum_{y \neq Y_i} R(X_i^{\mathcal{S}}, y)^2 \right) / |\mathcal{I}| + r_{cs},
\tag{9}
$$

where

$$
r_{cs} = \sum_{\{i_1, i_2\} \in \mathcal{I}^2, i_1 < i_2} \left( \prod_{X_{i_1}^{(j)} \neq X_{i_2}^{(j)}} (1 - p_j) \left( R(X_{i_1}^{\mathcal{S}}, Y_{i_1}) - R(X_{i_2}^{\mathcal{S}}, Y_{i_2}) \right)^2 \right).
\tag{10}
$$

In practice, $r_{cs}$ can be enforced batch-wise, and can be efficiently vectorized for the parallel computation of all $X_{i_1}^{(j)} \neq X_{i_2}^{(j)}$ pairs per batch using tensor operations. Note that since $R(X_i^{\mathcal{S}}, Y_i)$ are just model predictions, and $p_j$ are learned feature mask probabilities, each component in $E(X,Y)$ is easily accessible. When the labels are in the continuous space, the minimization objective with the consistency regularizer is derived the exact the same way to yield:

$$
E(X,Y) = \sum_{i \in \mathcal{I}} \left( Y_i - R(X_i^{\mathcal{S}}) \right)^2 / |\mathcal{I}| + r_{cs}.
$$

Our analysis is done with random variables $X$ and $Y$ to apply tools from probability theory. The data samples $\boldsymbol{X}$ and labels $\boldsymbol{Y}$ can be thought of as samples drawn from the distributions to which $X$ and $Y$ belong, where in the limit with infinitely many samples $\boldsymbol{X}$ and $\boldsymbol{Y}$ perfectly reflect these distributions.

## 4.4 SLM Computational complexity

As above, let $h$ be the hidden dimension, $n$ denote the number of samples, $b$ the batch size, and $N$ the total number of train steps; let $F_0$ be the total number of features, and $F_N$ the target number of features. We first discuss the complexity of individual components. The sparsemax operation is dominated by sorting, and hence has complexity $O(F_0 \log F_0)$ per sample, with an overall complexity of $O(nF_0 \log F_0)$. The consistency regularizer $r_{cs}$ in the MI-maximizing objective $E(\boldsymbol{X}, \boldsymbol{Y})$ has complexity $O(nbF_N)$, as the calculation $\prod_{\boldsymbol{X}_{i_1}^{(j)} \neq \boldsymbol{X}_{i_2}^{(j)}} (1 - p_j) \left( R(\boldsymbol{X}_{i_1}^{\mathcal{S}}, \boldsymbol{Y}_{i_1}) - R(\boldsymbol{X}_{i_2}^{\mathcal{S}}, \boldsymbol{Y}_{i_2}) \right)^2$ in Eq 10 occurs over the selected feature index set $j \in \mathcal{S}$, and is done between each sample and others in its batch. The non-regularizer component in $E(\boldsymbol{X}, \boldsymbol{Y})$ has complexity $nc$, where $c$ is the constant for the number of discrete or binned labels. Assuming an MLP classifier with $h$ hidden units, which has complexity $O(nh^2)$, the overall algorithm has complexity $O(nF_0 \log F_0 + nbF_N + nc + nh^2)$, making SLM amenable to scaling to a large number of features. In addition, SLM *amortizes* the cost of feature selection across batches throughout training, making it more scalable with respect to the number of samples. This is in contrast to PFA (Lu et al., 2007a) or many other MI-based methods such as CMIM (Fleuret, 2004) or JMIM (Bennasar et al., 2015), which place the memory and compute burden of selection for the entire dataset in the same step.

## 5 Experiments

### 5.1 Datasets and Settings

We present the efficacy of SLM in feature selection on wide range of datasets from numerous domains. For all experiments, we ensure fair comparison by employing similar hyperparameter search space and budget – to search for hyperparameters such as batch size and learning rate for each baseline method and dataset, we conduct an extensive random search within the search grid, by randomly generating a value within a conceivable range. We run a total of 300 trials for each method-dataset combination to ensure sufficient coverage, and tune all hyperparameters based on the validation accuracy. This process is chosen as it closely resembles model benchmarking and selection in real-world applications. The Appendix includes a myriad of additional experiments: selected feature interpretability (§A.6), compute timings (§A.7) and synthetic data experiments (§A.9) to demonstrate SLM's scalability, using the Hilbert-Schmidt Independence Criterion (HSIC) in lieu of the MI regularizer to demonstrate the effectiveness of the learned sparse mask (§A.8), as well as comparisons with further end-to-end baselines (§A.10).

We benchmark on a variety of real-world datasets across many domains, including computer vision, biological data, financial data, etc. Concretely, we benchmark on Mice, MNIST, Fashion-MNIST, Isolet, Coil-20, Activity, Ames Housing, and IEEE-CIS Fraud datasets. We use a 70-10-20 train/validation/test split; and when available, we use the exact same train/validation/test samples as (Lemhadri et al., 2019) for fair comparison. We give further detailed descriptions in §A.1. Cross entropy is used as the optimization objective for classification tasks, and MAE is used as the optimization objective for regression.

We benchmark SLM against a variety of competitive methods. The mutual information (**MI**) based feature selection baseline uses entropy estimation from k-nearest neighbors distances as described in (Kraskov et al., 2004; Ross, 2014) to estimate MI. Tree-based methods yield Gini importance scores, which can be used for feature selection. For this we benchmark two commonly used methods: random forest (**RF**) (Breiman, 2001), an ensemble of independent trees, and **XGBoost** (Chen & Guestrin, 2016), a scalable end-to-end tree boosting system. We furthermore benchmark against methods as discussed in §2: **LassoNet** (Lemhadri et al., 2019), which uses residual connections to allow the network to learn whether to use any given feature in a particular layer; feature importance ranking based on the **Fischer** score (Gu et al., 2012); principal feature analysis (**PFA**) (Lu et al., 2007a), a PCA-based method; and **HSIC-Lasso** (Yamada et al., 2014), which uses kernel learning to find non-linear feature interactions. Lastly, we benchmark against **linear** regression, where feature importance is determined by the learned feature coefficients. When available, we use results from (Lemhadri et al., 2019). For consistency and fairness, each baseline method uses the same input as SLM to select features, which are then passed to an MLP to compute the task metric.

## 5.2 Task performance with feature selection

| Method | Mice↑ | MNIST↑ | Fashion↑ | Isolet↑ | Coil-20↑ | Activity↑ | Ames↓ | Fraud↑ |
|---|---|---|---|---|---|---|---|---|
| All-features | 0.990 | 0.928 | 0.833 | 0.953 | 0.996 | 0.956 | 0.283 | 0.782 |
| **SLM (ours)** | 0.981 | **0.953** | **0.835** | **0.919** | **0.996** | **0.947** | **0.274** | **0.911** |
| Fisher | 0.944 | 0.813 | 0.671 | 0.793 | 0.986 | 0.769 | 0.286 | 0.743 |
| HSIC-Lasso | 0.958 | 0.870 | 0.785 | 0.877 | 0.972 | 0.829 | **0.273** | 0.846 |
| PFA | 0.939 | 0.873 | 0.793 | 0.863 | 0.975 | 0.779 | 0.356 | 0.852 |
| XGBoost | 0.968 | 0.913 | 0.832 | 0.879 | 0.986 | 0.926 | 0.403 | 0.872 |
| MI | 0.949 | 0.882 | 0.645 | 0.751 | 0.976 | 0.883 | 0.335 | 0.711 |
| Linear | 0.982 | 0.452 | 0.787 | 0.760 | 0.983 | 0.914 | 0.318 | 0.871 |
| Anova | **0.995** | 0.113 | 0.719 | 0.811 | 0.986 | 0.901 | 0.358 | 0.744 |
| RF | 0.967 | 0.928 | 0.829 | 0.892 | 0.993 | 0.893 | 0.303 | 0.773 |
| Lassonet | 0.958 | 0.873 | 0.800 | 0.885 | 0.991 | 0.849 | 0.342 | 0.842 |

Table 1: Test performance on real-world benchmarks with 50 selected features. SLM outperforms competitive baselines. The metrics reported are AUC for Fraud, since there is a high class imbalance; hence AUC is reported; the median MAE on standard-normalized labels is reported for Ames; and accuracy is reported on all other datasets. The arrow next to each dataset indicates whether a higher or lower value is more optimal. These test results are selected based on the best *validation set performance* during 300 hyperparameter grid search trials.

There is a rich body of work that assign importance to individual features (Lundberg & Lee, 2017; Shrikumar et al., 2017; Zintgraf et al., 2017; Chang et al., 2019). We consider feature importance to be measured by contribution towards the task metric, as accurate predictor performance is typically the end goal, and the importance of each individual feature is not always well-defined due to feature interactions. Therefore, we focus on benchmarking task predictive accuracy given the selected features as the metric.

First, we study selecting a fixed number of features across a wide range of high dimensional datasets (most with >400 features) and feature selection methods. We consistently choose 50 selected features, as this represents a small fraction of the total features for most datasets, as often done in practice. This number is kept consistent without tuning for any given method, to avoid favoring any given one. Table 1 shows that the SLM consistently yields competitive performance, outperforming all methods in all cases except on Mice and Ames, for which the performance is saturated due to small numbers of original features, making feature selection less relevant. Most feature selection methods are not consistent in their performance. On the other hand, SLM's strong performance is consistent – across a wide range of datasets, SLM selects the features accurately. Interestingly, we observe that there are cases where SLM even outperforms the baseline of using all features, which can likely be attributed to superior generalization when the limited model capacity is focused on the most salient features. Especially for datasets that are non i.i.d. in nature, feature selection can be a strong inductive bias integrated in the architecture, that can yield superior generalization.

| Method | Test AUC on Fraud Detection ↑ | | |
|---|---|---|---|
| | 20 features | 50 features | 100 features |
| **SLM (ours)** | **89.32** | **91.06** | **91.75** |
| Anova | 71.81 | 74.41 | 82.91 |
| RF | 72.16 | 77.29 | 78.72 |
| Linear | 84.32 | 87.11 | 87.46 |
| MI | 65.37 | 71.05 | 74.91 |
| XGBoost | 85.45 | 87.24 | 81.67 |

Table 2: Test AUC on the Fraud Detection dataset (Kaggle, 2022) at different feature selection levels: 20, 50 or 100 features selected. The superiority of SLM persists across different numbers of selected features.

Next, we conduct further experiments on the Fraud Detection dataset (Kaggle, 2022), a large-scale dataset with many heterogeneous features. It is highly non i.i.d. (Grover et al., 2022), thus making feature selection important given that high capacity models can be prone to overfitting and poor generalization. Table 2 shows that SLM outperforms other methods consistently for different number of selected features, and its performance degradation with respect to reducing the number of features is much smaller. Indeed, the AUC with 20 features out of 432, is >10% better than using all features, indicating improved generalization.

### 5.3  Ablation studies

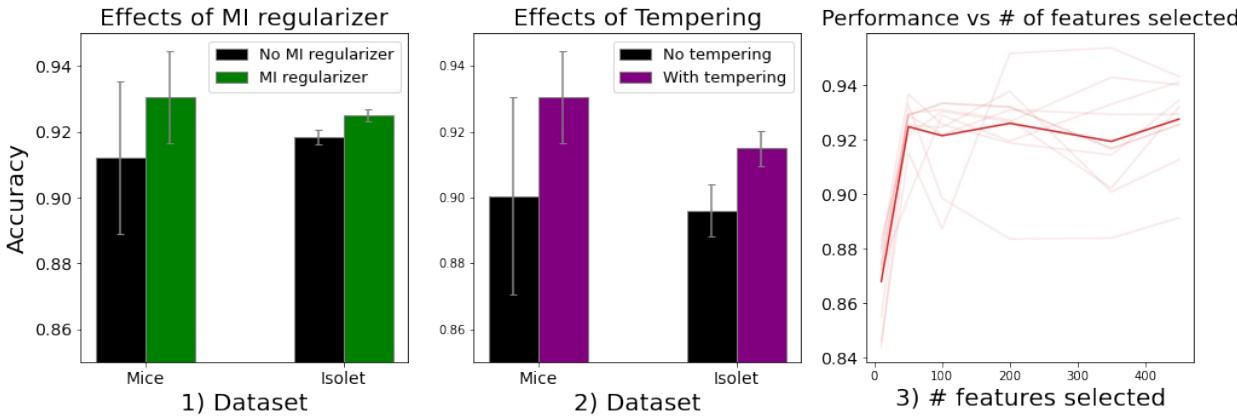

Figure 2: (1) and (2) show ablation studies on the effect of MI regularization and tempering the number of features. Both ablation studies have the same number (50) of selected features on all datasets. (3) shows the task accuracy as a function of the number of features selected on the activity dataset. The dark line shows the average of ten *random hyperparameter* trials, shown with light hue, demonstrating that task performance can be near-optimal even with a small subset of features.

We study the utility of SLM components, particularly the effects of the MI regularizer and tempering the number of features, which gradually decreases the number of selected features from the full feature set to the target number. The effects are measured by randomly selecting ten hyperparameter settings and a seed, and recording the average performance with or without either MI regularizer or tempering (without tempering refers to keeping the number of selected features constant throughout training.). Fig 2 shows that both MI regularization and tempering positively affect task performance. This is consistent with the theory developed in §3: the MI regularizer encourages maximal mutual information sharing between the labels and the selected features; and tempering allows the model to initialize learning based on all features, rather than a randomly selected subset.

## 6  Discussion

**Feature importance interpretability**. SLM learns a sparse mask $\mathcal{M}$ that contains the feature selection coefficients. We show that this approach yields superior results with end-to-end learning by allowing a smooth transition between selecting and un-selecting features. In addition, SLM can also be used for interpretation of global feature importance during inference, yielding the importance ranking of selected features, similar to other commonly-used methods like SHAP (Lundberg & Lee, 2017). This can be highly desired in high-stakes applications such as healthcare or finance, where an importance score can be more useful than simply whether a feature is selected or not.

**Feature interdependence during selection**. Compared to prior MI-based feature selectors (Ding & Peng, 2005; Fleuret, 2004; Bennasar et al., 2015), SLM accounts for feature inter-dependence by learning inter-dependent probabilities $\{p_j\}_j$ for the selected feature, where $\{p_j\}_j$ jointly maximize the MI between features and labels. Furthermore, SLM learns feature selection and the task objective in an end-to-end way, which alleviates the selection of repetitive features that may individually be predictive, as gradient descent

favors increasing the probability for a non-redundant and loss decreasing but less predictive feature over an individually predictive but redundant feature.

**Improved model generalization via feature selection**. Feature selection can help improve generalization beyond the training set, especially for high capacity models like deep neural networks, which can easily overfit patterns from spurious features that do not hold across training and test data splits (Arjovsky et al., 2019). For instance, Table 1 shows that on some datasets, especially with SLM, prediction on a subset of features can outperform that on all features. Furthermore, Fig 2 shows that task performance can reach near-optimum with even a small subset of all features. Therefore, feature selection is a potential alternative for alleviating compute cost during training and inference, without sacrificing on accuracy.

**Relation to other MI estimations in deep learning.** MI-based objectives have been used in other deep learning methods, such as InfoNCE (Oord et al., 2018), InfoGAN (Chen et al., 2016), and Deep Graph Infomax (Velickovic et al., 2019). To estimate MI, these typically train classifiers on samples drawn from the joint distribution and the product of the marginals, whose exact distributions can be intractable. In contrast, for feature selection, while the exact distributions of the features and the labels are known, the computation of their mutual information and its maximization is computationally intractable. To address this, SLM proposes a quadratic relaxation of MI optimization, applied to feature selection by converting MI maximization to minimizing a loss function. SLM does not need to sample from the joint or marginal distributions, a potentially computationally intensive process. Furthermore, prior works (Chen et al., 2016; Velickovic et al., 2019) often require a contrastive term in estimation of MI with negative sampling, a process that is not needed in SLM.

**Future work**. SLM can be integrated into unsupervised or semi-supervised learning, with modified objectives. In addition, our results indicate more significant outperformance for datasets with non i.i.d. characteristics as feature selection can effectively reduce the feature dimensionality and reduce the risk of overfitting to the spurious correlations of irrelevant features. Lastly, feature selection for data with structure (e.g. temporal or graph) is an interesting extension, which might be based on modifying SLM to apply masking to entire time-series or graph data.

## 7 Conclusion

We introduce SLM, a sparse learnable mask based feature selection framework that maximizes the MI between features and labels, while optimizing the training objective end-to-end. Learning the feature masks allows a smooth, probabilistic selection of features as well as insights on feature importance. SLM demonstrates competitive performance against SOTA baselines, and opens door to future applications in domains such as graph or time series representation learning.

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

# A   Appendix

## A.1   Dataset Details

This section provides additional details on the experimental data. We first consider the real-world benchmark datasets in (Lemhadri et al., 2019). **Mice** consists of protein expression levels measured in the cortex of normal and trisomic mice who had been exposed to different experimental conditions. Each feature is the expression level of one protein. **MNIST** and **Fashion-MNIST** consist of 28-by-28 grayscale images of hand-written digits and clothing items, respectively. The images are converted to tabular data by treating each pixel as a separate feature. **Isolet** consists of preprocessed speech data of people speaking the names of the letters in the English alphabet with each feature being one of the preprocessed quantities, including spectral coefficients and sonorant features. **Coil-20** consists of centered gray-scale images of 20 objects taken at certain pose intervals, hence the features are image pixels. **Activity** consists of sensor data collected from a smartphone mounted on subjects while they performed several activities such as walking or standing. For these datasets, we use the exact same data splits and preprocessing approaches with (Lemhadri et al., 2019) for fair comparison, as well as the same model hyperparameter search space.[1] In addition, we consider the **Ames** housing dataset (Cock, 2011), with the goal of predicting residential housing prices based on each home's features; as well as the IEEE-CIS **Fraud** Detection dataset (Kaggle, 2022), with the goal of identifying fraudulent transactions from numerous transaction and identity dependent features. Table 3 summarizes the characteristics of the datasets used in the experiments.

| Dataset | Number of samples | Number of features | Number of classes |
|---------|-------------------|--------------------|-------------------|
| Mice | 1080 | 77 | 8 |
| MNIST | 10000 | 784 | 10 |
| Fashion | 10000 | 784 | 10 |
| Isolet | 7797 | 617 | 26 |
| Coil-20 | 1440 | 400 | 20 |
| Activity | 5744 | 561 | 6 |
| Ames | 1460 | 81 | N/A |
| IEEE Fraud | 590540 | 681 | 2 |

Table 3: Attributes of datasets used in experiments.

## A.2   Experimental details

As described, we use hyperparameter tuning based on the validation accuracy for all cases. We use the Adam optimizer for training, with exponential decay. For benchmarks from (Lemhadri et al., 2019), for a fair comparison, our hyperparameter search space is same as the original paper. For Fraud, which is larger and more complex, we extend the search space as in Table 4.

| Hyperparameter | Search space |
|----------------|--------------|
| Batch size | [512, 1024, 2048, 4096] |
| Learning rate | [0.001, 0.003, 0.01] |
| Decay steps | [1000, 10000] |
| Decay rate | [0.7, 0.9, 0.95, 0.99] |
| Number of epochs | [30, 100, 200] |
| Number of hidden units | [50, 100, 200] |
| Number of layers | [1, 2, 3, 4] |

Table 4: Hyperparameter tuning search space for experiments on the Fraud dataset.

---

[1]We use a single layer multi-layer perceptron (MLP) as the predictor, where the number of units is chosen from $[M/3, 2M/3, M, 4M/3]$.

For baselines such as LassoNet, we tune additional method-specific hyperparameters. For instance, for LassoNet, in addition to the hyperparameters, we also tune the $\ell_2$ penalization on the skip connection, the hierarchy parameter, and the dropout rate. For XGBoost, we also tune the number of estimators and the maximum tree depth.

## A.3 Proof of Lemma 3.2

**Lemma 3.2.** Given a nonuniform vector $\boldsymbol{v} \in \mathbb{R}^K$, to obtain $F$ nonzero elements in sparsemax($\boldsymbol{v}$), $\boldsymbol{v}$ should be multiplied with the scalar

$$
m = \begin{cases} \left( \sum_{i=1}^{F+1} v_{(i)} - (F+1) * v_{(F+1)} \right)^{-1} & \text{if } |\text{sparsemax}(\boldsymbol{v}) > 0| > F \\ \left( \sum_{i=1}^{F} v_{(i)} - F * v_F \right)^{-1} & \text{if } |\text{sparsemax}(\boldsymbol{v}) > 0| < F, \end{cases} \tag{11}
$$

where $v_{(1)} \geq v_{(2)} \ldots \geq v_{(K)}$ denote sorted elements of $\boldsymbol{v}$ in descending order.

*Proof.* We first show the case when $|\text{sparsemax}(\boldsymbol{v}) > 0| > F$, i.e. the sparsity needs to be increased (the case where sparsity needs to be decreased works analogously). By (Martins & Astudillo, 2016), the projection of $\boldsymbol{v}$ onto $\Delta^{K-1}$ in Eq 1 takes the form sparsemax($\boldsymbol{v}$) $= [\boldsymbol{v} - \tau(\boldsymbol{v})]_+$, where $[x]_+ = \max\{0, x\}$, and $\tau$ takes the form $\tau = \frac{\left( \sum_{i \leq k(\boldsymbol{v})} v_{(i)} \right) - 1}{k(\boldsymbol{v})}$ with $k(\boldsymbol{v})$ defined as the index

$$
k(\boldsymbol{v}) := \max \left\{ k \in \{1, \ldots, K\} \mid 1 + k v_{(k)} > \sum_{i \leq k} v_{(i)} \right\}. \tag{12}
$$

Hence, increasing the sparsity such that sparsemax outputs only $F$ nonzero elements, i.e. decreasing the index $k(\boldsymbol{v})$ to $F$, requires finding the smallest $\boldsymbol{m}$ such that $1 + (F+1)\boldsymbol{m}v_{(F+1)} > \sum_{i \leq (F+1)} \boldsymbol{m}v_{(i)}$ does not hold, i.e. $F+1$ must be the first $k$ to fail the condition $1 + k v_{(k)} > \sum_{i \leq k} v_{(i)}$. Rewriting this condition in terms of $F$ we obtain:

$$
1 + (F+1)\boldsymbol{m}v_{(F+1)} > \sum_{i \leq (F+1)} \boldsymbol{m}v_{(i)}
$$
$$
\text{implies} \quad 1 > \boldsymbol{m} \left( \sum_{i \leq (F+1)} v_{(i)} - (F+1)v_{(F+1)} \right) \tag{13}
$$

The smallest $\boldsymbol{m}$ such that condition Eq. 13 does not hold is $\boldsymbol{m} = \left( \left( \sum_{i=1}^{F+1} v_{(i)} \right) - (F+1) * v_{(F+1)} \right)^{-1}$, which given Eq 12 implies $\boldsymbol{m}\boldsymbol{v}$ has $F$ nonzero elements. Analogously, to derive the multiplier for $\boldsymbol{v}$ to decrease sparsemax($\boldsymbol{v}$) sparsity, we need to increase the index $k(\boldsymbol{v})$ to $F$. This requires finding the largest $\boldsymbol{m}$ such that $1 + F(\boldsymbol{m}v_F) > \sum_{i \leq F} \boldsymbol{m}v_{(i)}$ holds, which implies: $\boldsymbol{m} = \left( \sum_{i=1}^{F} v_{(i)} - F * v_{(F)} \right)^{-1}$. $\qquad \square$

## A.4 Proof of Theorem 4.1

**Theorem 4.1.** Let $X$ and $Y$ denote the random variables representing the features and labels, respectively, and $\mathcal{Y}$ the value space for $Y$, then minimizing the optimum error $E(X, Y)$ in the model space $\{f : X \to Y\}$ is equivalent to maximizing the quadratic relaxation of mutual information $I_q(X, Y)$. More specifically,

$$
\min_{f:\mathcal{X} \to \mathcal{Y}} E(X, Y) = 1 - \sum_{y \in \mathcal{Y}} P_Y(y)^2 - I_q(X, Y)
$$

*Proof.* During training, the model seeks to produce the optimal predictions $R(x, y)$ that *minimize* $E(X, Y)$, while satisfying the constraint $\sum_{y \in \mathcal{Y}} R(x, y) = 1$. Hence we can apply Lagrange multipliers to solve for the optimal $R(x, y)$. Taking the derivatives of $E(X, Y)$ and the constraint $g(X, Y) = \sum_{x \in \mathcal{X}, y \in \mathcal{Y}} R(x, y) - |\mathcal{X}|$ with respect to $R(x, y)$:

$$
E'(X, Y) = \sum_{x \in \mathcal{X}, y \in \mathcal{Y}} -2P_{X,Y}(x, y) + 2P_X(x)R(x, y) \tag{14}
$$
$$
g'(X, Y) = \sum_{x \in \mathcal{X}, y \in \mathcal{Y}} 1
$$

Marginalizing $E'(X, Y)$ over $Y$ yields:

$$E'(X, Y) = \sum_{x \in \mathcal{X}} -2P_X(x) + 2P_X(x) = 0 \quad \vartriangleleft \text{Since } \sum_{y \in \mathcal{Y}} R(x, y) = 1$$

By Lagrange multiplier theory, for an optimum set of model predictions $R^*(X, Y)$, there exists some $\lambda$ such that $E'(X, Y)|_{R(X,Y)=R^*(X,Y)} = \lambda g'(X, Y)$. Since $E'(X, Y)|_{R(X,Y)=R^*(X,Y)} = 0$, $\lambda = 0$.

Therefore, by Eq 14, $R^*(x, y) = P_{X,Y}(x, y)/P_X(x)$. Plugging this into Eq 7, we obtain an expression relating the mutual information $I_q(X, Y)$ and the optimum error $E(X, Y)$:

$$\min_{f:\mathcal{X} \to \mathcal{Y}} E(X, Y) = 1 - 2 \sum_{x \in \mathcal{X}, y \in \mathcal{Y}} P_{X,Y}(x, y) R^*(x, y) + \sum_{x \in \mathcal{X}, y \in \mathcal{Y}} P_X(x) R^*(x, y)^2$$

$$= 1 - 2 \sum_{x \in \mathcal{X}, y \in \mathcal{Y}} P_{X,Y}(x, y) \frac{P_{X,Y}(x, y)}{P_X(x)} + \sum_{x \in \mathcal{X}, y \in \mathcal{Y}} \frac{P_{X,Y}(x, y)^2}{P_X(x)}$$

$$= 1 - \sum_{x \in \mathcal{X}, y \in \mathcal{Y}} \frac{P_{X,Y}(x, y)^2}{P_X(x)}$$

$$= 1 - \sum_{y \in \mathcal{Y}} P_Y(y)^2 - I_q(X, Y) \quad \vartriangleleft \text{By Eq 6}$$

Since $P_Y(y)$ is fixed for a given dataset, minimizing $E(X, Y)$ across the model space is equivalent to maximizing $I_q(X, Y)$. $\qquad \square$

**Continuous label space**. For the case with continuous labels (as occur for regression problems), the quadratic relaxation analogue of Eq 6 becomes:

$$\tilde{I}_q(X, Y) := \left( \sum_{x \in \mathcal{X}} \int_{y \in \mathcal{Y}} P_{X,Y}(x, y)^2 / P_X(x) \right) - \int_{y \in \mathcal{Y}} P_Y(y)^2. \tag{15}$$

Let $\mathcal{Y}_k$ be a discretization of the continuous labeling space $\mathcal{Y}$ with $k$ intervals, i.e. we are turning the continuous labeling function $\mathcal{X} \to \mathcal{Y}$ into a piecewise constant one with $k$ steps. Then, discretizing followed by taking the limit as $k \to \infty$ (and all bin sizes tend to 0) yields, analogous to Eq 7:

$$\tilde{E}(X, Y) := \lim_{k \to \infty} \sum_{x \in \mathcal{X}, y \in \mathcal{Y}_k} P_{X,Y}(x, y) \left( (1 - R(x, y))^2 + \sum_{y' \in \mathcal{Y}_k \setminus y} R(x, y')^2 \right)$$

$$= 1 - 2 \sum_{x \in \mathcal{X}} \int_{y \in \mathcal{Y}} P_{X,Y}(x, y) R(x, y) + \sum_{x \in \mathcal{X}} \int_{y' \in \mathcal{Y}} P_X(x) R(x, y')^2$$

Therefore following the same logic as in the proof above:

$$\min_{f:\mathcal{X} \to \mathcal{Y}} \tilde{E}(X, Y) = 1 - \sum_{x \in \mathcal{X}} \int_{y \in \mathcal{Y}} \frac{P_{X,Y}(x, y)^2}{P_X(x)}$$

Plugging Eq 15 into this equation thus gives:

$$\min_{f:\mathcal{X} \to \mathcal{Y}} \tilde{E}(X, Y) = 1 - \int_{y \in \mathcal{Y}} P_Y(y)^2 - \tilde{I}_q(X, Y),$$

as desired. Thus, minimizing the optimum objective $\tilde{E}(X, Y)$ across the model space $\{f : X \to Y\}$ is equivalent to maximizing $I_q(X, Y)$.

### A.5 Properties of sparsemax for SLM

This section further details the key aspects of utilizing sparsemax for SLM, in terms of how it compares with softmax with thresholding for feature selection, as well as how SLM avoids the sparsemax support collapse problem.

### A.5.1 Sparsemax vs softmax with thresholding

Softmax is the commonly used nonlinear normalization function. An alternative method for learning the sparse mask $\mathcal{M}_{sp}$ would be to apply softmax normalization, followed by a top-k operation, and an additional normalization to render it a probability mask. This method is not only unwieldy with additional steps, but because the softmax-top-k normalization normalizes with respect to the *absolute* value of $\boldsymbol{v}$, whereas sparsemax normalizes with respect to its *relative* values (by subtracting a $\boldsymbol{v}$-dependent threshold), sparsemax($\boldsymbol{v}$) is more equi-distributed over the interval $[0, 1]$ than softmax-top-k normalization (i.e. sparsemax($\boldsymbol{v}$) has lower entropy than softmax-top-k normalization), making it more discriminatory for feature selection. Furthermore, one gradient computation advantage of sparsemax($\boldsymbol{v}$) is that it allows a faster computation of the Jacobian-vector product – which typically suffices for backpropagation – in $O(S(\boldsymbol{v}))$ time, $S(\boldsymbol{v})$ being the support of $\boldsymbol{v}$ (Martins & Astudillo, 2016), compared with linear time (with respect to the size of $\boldsymbol{v}$) for softmax.

### A.5.2 SLM avoids sparsemax support collapse

In practice, since the gradient of sparsemax is zero for elements outside its support (Martins & Astudillo, 2016), an element that initially falls outside its support would stay so throughout training. This would lead to sparsemax support collapse, i.e. the size of the support dwindles during training.

SLM avoids the sparsemax support collapse problem, due to its sparsemax argument scaling (Lemma 3.2). In vanilla sparsemax, as the support of sparsemax is determined by the distances between the top-ranking elements, having non-zero gradients only for the elements in the support of sparsemax causes only those elements to drift. As the inputs to sparsemax can vary without bound, this drift becomes larger over time, i.e. the distance between the top elements, which are in sparsemax's support, becomes larger, making the sparsemax support smaller, and blocking new elements from entering the support. However, when the input to sparsemax is scaled as in Lemma 3.2, this drift is controlled, and can shorten the distances (in addition to lengthening them) between the top elements, hence the sparsemax support can acquire new elements, avoiding the collapse.

### A.5.3 Experimental analysis of the number of features in support

Experimentally, when training SLM feature selection with a single-layer MLP architecture, on a dataset of 1000 samples with 100 features each, using sparsemax *with* scaling to select 30 features consistently yields 30 features in the sparsemax support. In contrast, without scaling, the sparsemax support consistently dwindles to well below 10 features within 15 epochs.

### A.6 Feature Interpretability Results

While SLM optimizes feature selection for the task metric, the fact that the selected features are global readily opens the door for feature importance interpretability applications, as the chosen features can give insights about the task. To this end, we focus on the Ames housing dataset (Cock, 2011), as its features are easily understandable. As mentioned in §A.1, the features in the Ames dataset consist of characteristics of houses, and the prediction target is the house price. We use the model parameters found in the best validation trial reported in Table 1, and select the top ten out of the 81 features. To obtain importance scores of the selected features, we study the selection probabilities learned in the feature mask. Using this, the ten highest-probability features in terms of determining a house's prices are, with learned feature probabilities: 'OverallQual' (0.211), 'FullBath' (0.182), 'GarageCars' (0.124), 'BsmtFullBath' (0.0795), 'MSSubClass' (0.0758), 'GarageFinish' (0.0739), 'HalfBath' (0.0718), 'PoolArea' (0.0562), 'Fireplaces' (0.0473), 'HouseStyle' (0.0403).

Some aspects of this selection conform to common sense – the overall quality of the property, the number of bathrooms, and the size of the garage or pool are good predictors of housing value. Other aspects are more surprising, for instance the feature 'BedroomAbvGr' – the number of bedrooms above ground – is not selected, even though one would expect the number of bedrooms to be an important selling factor. However, on further thought, as the number of bedrooms is positively correlated with the number of bathrooms (Eggers & Moumen, 2013), SLM is avoiding feature redundancy by only selecting one of the correlated features. The

same reasoning applies for the features 'OverallQual', the overall quality, which is selected, and 'OverallCond', the overall condition, which is not selected.

## A.7 Computational Complexity Experiments

As stated in § 4.4, let $F_0$ be the total number of features, and $n$ the number of samples, SLM has $O(nF_0 \log F_0)$ dependence on $F_0$. To test that this low complexity in theory translates to actual fast feature selection in practice, we present the wall clock timing of SLM. We compare specifically against LassoNet, a strong baseline that also selects features end-to-end. Table 5 shows the timing results for one epoch on the Mice dataset, demonstrating that SLM's low complexity in theory also translates to fast execution in practice.

| Feature Selection Method | Timing (s) |
|---|---|
| SLM | **1.21** $\pm$ 0.016 |
| LassoNet | 19.62 $\pm$ 0.796 |

Table 5: Timing results for one epoch on the mice dataset between SLM and LassoNet, a strong baseline that also selects features end-to-end. This comparison is down under the exact same settings for both methods: hidden dimension of 64, batch size of 256, one MLP predictor layer, selecting 50 features, run on a single V100 GPU. The result statistics are collected over five different runs. Only the training component is measured, not including data splitting and processing.

Furthermore, we discuss the computation of the MI objective in Eq 9. In particular, the consistency term $r_{cs}$ in Eq 10, which ensures that if two samples have the same values in their selected features, their model predictions are the same as well. In theory, if we imagine giving $r_{cs}$ a weight coefficient $\alpha$ in Eq. 9, with the interpretation that in the limit where $\alpha \to \infty$, this consistency is strictly enforced; and in the limit where $\alpha \to 0$, not at all. In practice, given that they have the same orders on terms such as model predictions $R(X,Y)$ by design, $r_{cs}$ and the remaining term in Eq 9 have the same order of magnitude. Furthermore, $r_{cs}$ indeed is the most compute-intensive part in Eq 9, as $r_{cs}$ requires pairwise comparisons within the batch (for each pair $X_{i1}, X_{i2}$ it is computed over the feature indices $j$ where $X_{i1}^{(j)} \neq X_{i2}^{(j)}$). Experimentally, on the Ames dataset with a batch size of 128, the $r_{cs}$ computation takes up $1.51 \pm 0.012$ ms, out of $1.94 \pm 0.017$ ms for the entire MI regularizer computation in Eq 9. This reveals an interesting accuracy-compute trade-off, where users may want to skip the $r_{cs}$ computation for an even faster, approximate MI regularizer computation.

## A.8 HSIC objective to demonstrate the effectiveness of the learned sparse mask

While the mutual information regularizer is an integral part of the SLM, in this section we show the *effectiveness and generalizability* of the learned feature selection mask approach, by replacing the MI regularizer with another measure of dependency between random variables: the Hilbert-Schmidt Independence Criterion (HSIC) (Gretton et al., 2005). Analogous to how we apply the mutual information regularizer, we consider the HSIC between the features random variable and the labels random variable distributions.

Concretely, for two random variables $X$ and $Y$, the HSIC is the Hilbert-Schmidt norm of the covariance operator between these random variable distributions in the Reproducing Kernel Hilbert Space, defined as:

$$
\begin{aligned}
\text{HSIC}(\mathbb{P}_{XY}, \mathcal{F}, \mathcal{G}) &\coloneqq ||\text{Cov}(X,Y)||_{HS}^2 \\
&= \mathbb{E}_{XX'YY'}[k_X(X,X')k_{Y'}(Y,Y')] + \mathbb{E}_{XX'}[k_X(X,X')]\mathbb{E}_{YY'}[k_Y(Y,Y')] \\
&\quad - 2\mathbb{E}_{XY}[\mathbb{E}_{X'}[k_X(X,X')]\mathbb{E}_{Y'}[k_Y(Y,Y')]],
\end{aligned}
\tag{16}
$$

where $k_X$ and $k_Y$ denote kernel functions; $\mathcal{F}$ and $\mathcal{G}$ are the Hilbert spaces of functions on $X$ and $Y$; and $\text{Cov}(X,Y)$ denotes the cross-covariance operator $\mathcal{F} \to \mathcal{G}$ (Gretton et al., 2005). We experimented with different kernel functions for Eq. 16, and chose the Gaussian kernel based on final task performance: $k(\mathbf{x}, \mathbf{x}') \coloneqq \exp(-||\mathbf{x} - \mathbf{x}'||^2/(2\sigma^2))$, where $\mathbf{x}$, $\mathbf{x}'$ are samples drawn from the distribution $\mathbb{P}_X$, with $\sigma$ being the standard deviation.

Unlike mutual information, HSIC is not a probabilistic measure, and does not have an interpretation in terms of information theoretic quantities (bits or nats) (Ma et al., 2020). On the other hand, HSIC does

not require any probability density estimation, which can be a computational bottleneck in approximating mutual information. In our experiments, we compute the HSIC objective batch-wise between the features and the labels, conditioned on the learned feature selection mask. This conditioning is done by scaling the standard-normalized input features by the learned feature selection mask, before the mask is sparsified via sparsemax. Table 6 shows the results of our HSIC experiments. Overall, we observe the version of SLM with HSIC to be worse than the original version, but it does improve over the other baselines (and most notably, significantly better than HSIC-Lasso, a commonly-used feature selection method that integrates the HSIC objective as well).

| Feature Selection Method | Isolet | Activity |
|---|---|---|
| **SLM** | **0.919** | **0.947** |
| SLM with HSIC (instead of MI) | **0.918** | 0.931 |
| LassoNet | 0.885 | 0.849 |
| Fisher | 0.793 | 0.769 |
| HSIC-Lasso | 0.877 | 0.829 |
| PFA | 0.863 | 0.779 |
| XGBoost | 0.879 | 0.926 |
| MI | 0.751 | 0.883 |
| Linear | 0.760 | 0.914 |
| Anova | 0.811 | 0.901 |
| Random Forest | 0.892 | 0.893 |

Table 6: Feature selection performance, including replacing the MI regularizer in SLM with the HSIC objective. The hyperparameter tuning based on the validation set is performed similarly to the main experiments in §5.2. SLM learned feature mask combined with the HSIC objective perform very competitively compared to the myriad of strong baselines, demonstrating the effectiveness and generalizability of SLM's learn feature selection mask approach.

### A.9 Synthetic Data Experiments

We demonstrate the performance of SLM on a synthetic dataset that is specifically constructed such that only a small subset of features affect the output value while the vast majority are not useful for the task. All input features $\mathbf{X_{i,j}}$ are sampled from the uniform distribution $U[-1,1]$ and the noise at the end $\epsilon_{\mathbf{i,j}}$ are sampled from standard Gaussian random variable with zero mean and unit variance. The input-output relationship are governed by the equations shown below:

$$\mathbf{T_{i,j}^{(1)}} = \frac{1}{L} \sum_{i=1}^{L} \exp(\mathbf{X_{i,j}}), \tag{17}$$

$$\mathbf{T_{i,j}^{(2)}} = \exp(\frac{1}{L} \sum_{i=L+1}^{2L} |\sin(2\pi\mathbf{X_{i,j}}|), \tag{18}$$

$$\mathbf{T_{i,j}^{(3)}} = \frac{1}{L} \sum_{i=2L+1}^{3L} -\log(1.1 + \mathbf{X_{i,j}})), \tag{19}$$

$$\mathbf{T_{i,j}^{(4)}} = \frac{1}{L} \sum_{i=3L+1}^{4L} \mathbf{X_{i,j}}, \tag{20}$$

$$\mathbf{T_{i,j}^{(5)}} = 1/(1 + \frac{1}{L} \sum_{i=4L+1}^{5L} |\tanh(\mathbf{X_{i,j}})|), \tag{21}$$

$$\mathbf{Y_{i,j}} = \begin{cases} 1, & \text{if} \quad (\mathbf{T_{i,j}^{(1)}} + \mathbf{T_{i,j}^{(2)}} + \mathbf{T_{i,j}^{(3)}} + \mathbf{T_{i,j}^{(4)}} + \mathbf{T_{i,j}^{(5)}} - 3 + 0.2\epsilon_{i,j}) > 0, \\ 0, & \text{otherwise} \end{cases} \qquad . \qquad (22)$$

As can be seen, the function is highly nonlinear in dependence to the input features, and in total $5L$ features are salient.

| Hyperparameter | Search space |
|---|---|
| Batch size | [128, 256, 512] |
| Learning rate | [0.001, 0.003, 0.01] |
| Decay steps | [1000, 10000] |
| Decay rate | [0.7, 0.9, 0.95, 0.99] |
| Number of epochs | [30, 100, 200] |
| Number of hidden units | [30, 50, 100] |
| Number of layers | [1, 2, 3,] |

Table 7: Hyperparameter tuning search space for experiments on the Synthetic dataset.

We construct the dataset with 3000 features among which only 100 or 300 are salient, i.e. $L = 20$ or $L = 60$ for two different training dataset size values, 35000 and 14000 training samples respectively. Train-validation-test are split with 0.7-0.1-0.2 ratio, similar to all other experiments and hyperparameter tuning is done with the search space presented in 7. We compare SLM with other feature selection methods, when they are used to select the 300 features. Table 8 and 9 highlight the superior performance of SLM compared to the alternative methods for challenging datasets with a very large number of features.

| Feature selection method | Test accuracy (%) |
|---|---|
| SLM | **71.5** |
| Anova | 67.9 |
| RF | 62.6 |
| Linear | 67.0 |
| MI | 63.0 |

Table 8: Test accuracy (%) on the Synthetic dataset with 300 salient features ($L = 60$) and 14000 training samples.

| Feature selection method | Test accuracy (%) |
|---|---|
| SLM | **73.9** |
| Anova | 69.0 |
| RF | 69.9 |
| Linear | 69.7 |
| MI | 61.2 |

Table 9: Test accuracy (%) on the Synthetic dataset with 100 salient features ($L = 20$) and 35000 training samples.

### A.10    Further Comparison with End-to-end Baselines

One of SLM's strengths is end-to-end feature selection along with task learning, which allows the model to incorporate inductive biases from the task directly into feature selection. Therefore, we specifically focus on comparing SLM with additional end-to-end feature selection methods, beyond the results in Table 1. As discussed in §2, Concrete Autoencoder (Abid et al., 2019) proposes an *unsupervised* feature selector based on using a concrete selector layer as the encoder and using a deep neural network as the decoder. FsNet (Singh et al., 2020) uses a concrete random variable for discrete feature selection in a selector layer and a supervised

deep neural network regularized with the reconstruction loss, with a focus on biological data, which are often high-dimensional with limited sample size. STG (Yamada et al., 2020) develops a fully embedded supervised method that learns stochastic gates with a probabilistic relaxation of the count of the number of selected features. While all these works selects features and learns task prediction end-to-end, given that SLM is a supervised model, with a general focus beyond the high-dimensionality and low-sample-size setting, STG (Yamada et al., 2020) is the strongest, most related baseline to compare SLM with. Table 10 shows the comparison between SLM and STG on the Isolet and Activity datasets with 50 selected features. There are certain similarities between how SLM and STG control which feature to select: SLM learns a sparse probability mask $m$ for the features, whereas STG learns learn the parameters of the approximate Bernoulli distributions via gradient descent for each feature. While STG learns the parameters for each Bernoulli variable independently, one advantage SLM has is accounting for interdependence amongst selected features, through both the fact that the probabilities in $m$ are interdependent, and through the MI regularizer (further details discussed in §6).

| Feature Selection Method | Isolet ↑ | Activity ↑ |
|---|---|---|
| SLM | **92.49** $\pm$ 0.20 | **93.35** $\pm$ 0.82 |
| STG | 84.50 $\pm$ 1.98 | 91.81 $\pm$ 0.71 |

Table 10: Test accuracy (%) comparison between SLM with a closely related, end-to-end feature selection baseline STG, which controls feature selection via learned stochastic gates, on the Isolet and Activity datasets with 50 features selected. The two methods are compared under the exact same conditions to the largest extent possible: using the same hidden dimension, number of epochs, batch size, learning rate, etc., all randomly generated from within a feasible range. The non-shared hyperparameters are also generated from random within a feasible range. The results are averaged over ten different runs. SLM is able to account for interdependence amongst selected features, through the learned mask $m$ and the MI regularizer.

