# OpenReview forum: "SLM: End-to-end Feature Selection via Sparse Learnable Masks"
_TMLR — Rejected by TMLR_

### Review · Reviewer_kDMo · 2023-02-23

**Summary Of Contributions:**

This paper proposes an end-to-end feature selection mechanism through sparse learnable masks. The authors use a new objective function that aims to maximize the mutual information (MI) between the selected features and the labels, which is derived from a quadratic relaxation of mutual information. Experiments on some benchmark datasets have been conducted.

**Audience:**

Yes

**Claims And Evidence:**

No

**Requested Changes:**

Please adjust the submission according to the previous comments.

**Strengths And Weaknesses:**

** Strength **
1. The paper is relatively well-written. It is easy to follow.
2. The idea of using a differentiable learnable mask for end-to-end feature selection is interesting.

** Weakness **
1. The authors claimed that one of the advantages of using learnable mask (which zeros out non-informative features) is that the resulting feature selection is more interpretable. However, I do not see in the paper how the feature mask (and the selected feature) look like after training.
2. The statement in the first paragraph on the third page about the advantage of global feature selection over local feature selection seems misleading. Why global selection provides robustness benefit  when there is distribution shift between training and testing datasets? Isn't supposed to be opposite?
3. In algorithm 1, why the training task loss and MI loss need to be both calculated? Is that redundant?
4. Theorem 3.3 seems unlikely to be correct. I might have misunderstood the notations, but isn't R(x, y), i.e., the model's probability output for sample x and outcome y, dependent on the model? How is it possible that the loss function E(X, Y) defined based on R(X, Y) can be equivalently calculated using I_q(X, Y), which does not depend on the model?
5. Simply saying equation 7 can be generalized to continuous (regression) case through quantization is not helpful at all. One might get too large of a value space Y after quantization.

---

> ### Author Response · Authors · 2023-02-27
> **Thank you for your review, we have further clarified the points you made in the paper.**
>
> Thank you for the thoughtful review of our work. Please find our responses below:
>
> 1. The learnable mask gives the interpretable insights on which features are used by the model. Essentially, the features that have the mask coefficient of 0 are not selected, i.e. they are not used by the model. The methods section (Section 3) describes the form and use of the learned sparse mask vector $\mathbf{m_{sp}} \in \mathbb{R}^d$, which zeros out non-selected features via an element-wise product: $\mathbf{X} \odot \mathbf{m_{sp}}$, as mentioned. In Appendix A.6, we exemplify the selected features on a real-world dataset and explain how they conform to the common sense for that task.
>
> 2. Deep neural networks are prone to overfitting and poor generalization when the input is high dimensional with many redundant features, and this gets more severe as the test distribution deviates from the training distribution. Global feature selection reduces the amount of redundant features based on the entire dataset (that can be thought of as a global inductive bias), and hence improves the generalization of the predictor trained on such features (as shown in Table 2 for the real-world Fraud Detection dataset). In particular, under train-test distribution shift, local feature selection would cause the model to more likely pick up spurious features at test time, as the model has not seen the test distribution during training. Whereas for global feature selection, this only happens when a relevant feature during training becomes spurious in the test data. Global feature selection can also be thought of as a stronger inductive bias, constraining the model capacity in a judicious way.
>
> 3. By the training task loss, we refer to the objective function of optimizing for the dataset labels $\{y\}$. But since we are also performing feature selection in an end-to-end way and the search space for which features to select is very large, we want to add additional inductive bias to the model via the MI objective (defined in Eq. 9), which is conditioned on the feature selection mask. We have updated the term “training task loss” to “dataset task loss” to make this more clear.
>
> 4. We can see where you are coming from. The claim in Theorem 3.3 is that any model in the model function space that minimizes the objective $E(X, Y)$, does so effectively by maximizing $I_q(X, Y)$ (which can be maximized as different features are selected). I.e. the _optimal_ $R(X, Y)$ achievable in the model space (can be infinite dimensional) is obtained by maximizing $I_q(X, Y)$. By taking this optimal $R(X, Y)$ across all possible models, we make _minimizing_ $E(X, Y)$ independent of any particular model. We have updated the statement of Theorem 3.3 to make this more clear.
>
> In further detail, we are leveraging the connection between the formulations of this particular quadratic objective $E(X, Y)$ (defined in Equation set 7) and the mutual information quadratic relaxation $I_q(X, Y)$ to make minimizing $E(X, Y)$ and maximizing $I_q(X, Y)$ equivalent, as shown in the proof of Theorem 3.3.
>
> 5. The statement refers to first discretizing the continuous label space, and then taking the limit as the discretization becomes infinitesimal. When taking this limit, the discrete sums over $\mathcal{Y}$ become integrals over $\mathcal{Y}$ (the integrals converge, as the probabilities integrate to 1 over the state space) [Principles of Mathematical Analysis, Rudin 1953]. We have added more details on this in Appendix A.4, immediately following the proof of Theorem 3.3. Thank you for pointing this out.
>
> We'd be very happy to address any further points.

---

> > ### Comment · Reviewer_kDMo · 2023-04-03
> > **Post-rebuttal comment**
> >
> > After reading through the authors' response, the revised paper and the other reviews, I still cannot recommend acceptance of the paper. My concerns of the paper come mainly from its clarity and technical correctness.
> >
> > 1. The quadratic approximation of the mutual information requires $P_{X, Y}(x, y)/ P_X(x)$ and $P_Y(y)$ to be in a delta neighborhood of 1. However, it is not clear to me why this assumption holds. In fact, I would say this assumption generally does not hold.
> > 2. The notation $E(X, Y)$ is still confusing and needs much more clarification. From Theorem 4.1, it appears that E(X,Y) should depend on the model $f:X\to Y$. From Equation (8), it appears that $E(X, Y)$ also depend on the index set $S$ on the feature space.
> > 3. The authors say that extension from discrete space to continuous space is trivial through quantization. However, this is not that simple, especially when the input and/or output is high-dimensional. In particular, it is not clear to me how the authors can implement the loss $r_{cs}$ in the continuous case.
> >
> > The paper seems to be well-motivated and clearly explained until it reaches the technical part (section 4). I think the paper could still use another major revision to include a better explanation of the limitation (discrete vs continuous) of the algorithm and a clearer presentation of the notations.

---

> > > ### Author Response · Authors · 2023-04-04
> > > **We would like to clarify some misunderstandings.**
> > >
> > > We would like to clarify some misunderstandings:
> > >
> > > 1. As described in Section 4.1, the closeness of the approximation $I_q(X, Y)$ to $I(X, Y)$ relies on the fact that both $p \log q$ and $pq$ are convex with respect to $p$ and $q$, and hence exhibit the *same correlation* behavior with respect to $p$ and $q$, which is key to make this approximation work in practice. From an optimization perspective (i.e. in terms of closeness of gradients), this approximation works the best in a delta neighborhood of 1. The fact that $I_q(X, Y)$ is a good approximation of $I(X, Y)$ in practice is demonstrated in the experimental results.
> > >
> > > 2. As described in the paper, as well as in point 4 in the above response, while any given instance of the quadratic error term $E(X, Y)$ depends on the model that gives rise to that error, the *minimal* $E(X, Y)$, optimized across the model space $f: X \to Y$, is *independent* of any particular model. I.e. the minimal achievable $E(X, Y)$ does not depend on any particular model. This independence is consistent with the fact that the right hand side of the equation in Theorem 4.1 is independent of any model.
> > >
> > > Note that the fact that any particular instance of $E(X, Y)$ depending on the model is consistent with its dependence on the index set $S$ in Eq. 8, as the selected feature set $S$ is learned by the model.
> > >
> > > 3. As described by the paper (page 7), the loss in the continuous case is given exactly by the equation at the end of page 7. In particular, the regularization term $r_{cs}$ is implemented as in Eq. 10 in both the continuous and the discrete case, where we take the product over the difference in predictions (the $R$ terms), weighted by 1 minus the probabilities $p_j$ as given by the learned mask.
> > >
> > > Furthermore, we do not claim that the extension from the discrete to the continuous case is trivial. As described in the paper and point 5 in the reply above, we provide justification for focusing on the discrete case, and extension to the continuous case, in Section 4. Indeed, there is a long line of research on mutual information and entropy estimation that focuses on the case where the random variables live in the discrete space [Paninski 2003, Kraskov et al 2004, Valiant and Valiant 2011, Han et al 2015, Jiao et al 2015, Wu and Yang 2016]. This is because many variables in machine learning are indeed discrete, e.g. vocabulary indices for text data used in NLP, categorical variables such as whether transaction is fraudulent, etc. Furthermore, we are building on well-established literature that MI estimation in the continuous case can be reduced to the discrete case via binning and taking a limit [Kraskov et al 2004, Paninski 2003], as done in Sec. A.4.
> > >
> > > References:
> > >
> > > Alexander Kraskov, Harald Stögbauer, and Peter Grassberger, Estimating mutual information. PHYSICAL REVIEW E 69, 066138 (2004).
> > >
> > > Liam Paninski. Estimation of Entropy and Mutual Information. Neural Computation, 15, 1191–1253 (2003).
> > >
> > > Yanjun Han, Jiantao Jiao, and Tsachy Weissman. Adaptive estimation of Shannon entropy. In Information Theory (ISIT), pages 1372–1376. IEEE, 2015.
> > >
> > > Jiantao Jiao, Kartik Venkat, Yanjun Han, and Tsachy Weissman. Minimax estimation of functionals of discrete distributions. IEEE Transactions on Information Theory, 61(5):2835– 2885, 2015.
> > >
> > > Gregory Valiant and Paul Valiant. Estimating the unseen: an $n/\log (n)$-sample estimator for entropy and support size, shown optimal via new CLTs. In Proceedings of the forty-third annual ACM symposium on Theory of computing, pages 685–694. ACM, 2011.
> > >
> > > Yihong Wu and Pengkun Yang. Minimax rates of entropy estimation on large alphabets via best polynomial approximation. IEEE Transactions on Information Theory, 62(6):3702–3720, 2016.
> > >
> > > Thank you again for your review. We hope you take these clarifications into account.

---

### Review · Reviewer_24kh · 2023-02-27

**Summary Of Contributions:**

This paper studied the feature selection problem, in which the most relevant features are learned using a differentiable sparse mask. Notably, the feature map is discovered via quadratic loss, which is related to mutual information between the feature and the output. Experiments are carried out on a variety of tabular and simple image datasets (such as MNIST).


**Audience:**

Yes

**Claims And Evidence:**

No

**Requested Changes:**

The proposed concept, in my opinion, is worthy publication of TMLR. However, there are significant issues with the presentation and clarity (particularly in Sec 3). As a result, I would advise authors to address the aforementioned clarity and technical issues.


**Strengths And Weaknesses:**

### Review Summary
This paper's motivation and goal are clear and interesting: feature selection using a sparse and differentiable mask. Experiments on a variety of tabular datasets and simple image datasets demonstrate its utility. However, there are several significant limitations, including the paper's clarity, presentation, and critical technical concerns. As a result, I think the current version requires **major** revisions.

### Would some individuals in TMLR's audience be interested in the findings of this paper?

Yes. Feature selection is fundamental and crucial for machine learning and recent explainable AI. I believe the contribution is meaningful.

### Are the claims made in the submission supported by accurate, convincing and clear evidence?

Partially. I suggest major revisions. The main points are as follows:

(a) **Paper clarity and presentation.**

- Overall structure. The proposed method is not clearly introduced in this paper. Algo.1 provided an overall method description. However, details of Algo.1 **last 4 pages**, making this reviewer difficult to follow and back-checking algorithm 1 for long periods of time. As a result, I would suggest rewriting this section to include (1) clear and detailed method descriptions in Algo 1 and (2) concise discussions of each component in Sec 3.

- Math notations. The authors introduce multiple notations without explanation in Section 3. For example, Eq(2), sparsemax could be expressed more clearly. Figure 1 is difficult for me to understand, so I could suggest a better illustration. Lemmas 3.2 and (4) added more notations without clearly illustrating them.

- Mutual Information. I have concerns on the definition of mutual information in Eq(5). In general, in classification/regression problems, the input $X$ should be a continuous random variable and $Y$ could be continuous (in regression) and discrete (in classification). Thus the whole operation of sum on X seems problematic, implying that X could only task possibly countable values, which is not true in machine learning. The input to machine learning is typically assumed to be a continuous random variable. As a result, eq(5,6,7) and theorem 3.3 appear to be problematic in machine learning.

- Concerning the prediction loss $E(x,y)$. I'm not sure what to make of the form $E(x,y)$; it appears to be an energy term, right? Why do we need to consider the following form
$$ (1-R(x,y))^2 + \sum_{y^{\prime}} R(x,y^{\prime})^2$$

Why do we need to think about this specific term? What is the reason for this loss? After some thought, I realised that $y=0,1$ is equivalent to $2(1-R(x,y))2$. However, because y is continuous in the regression settings, how can we estimate the sum $\sum y^\prime R(x,y^\prime)^2$? This, I believe, is a problem in this case.

- The Mask sparsity in Sec 3.1. I think this part requires further discussions. Personally I think softmax+temperature is sufficient. There is no need to compute the projection part. Specifically:
Softmax+temperature = softmax(x/tem)

This could easily control the sparse level. Moreover, this is computational efficient. Could you tell me the computational complexity of computing projection through sparemask?

I noticed the discussion in A5, while I think softmax+temperature could do a similar task (if we choose a proper temperature). This is also required in sparse-mask, a scaling value is required.

As for the gradient-computation, the challenge in spare-mask is to solve the optimization objective in EQ(4).
Still in Eq(4), why not a L1 norm constraint for directly obtaining a projection? This could avoid scaling, right? Overall, all of these require further discussions/justifications/analysis.

- About $I_q(X,Y)$ in equation (6), I would think it is the mutual information through a f-divergence, **rather than approximations of KL divergence ** (This claim is seemingly incorrect). I suggest paper [1] f-divergence Inequalities(https://arxiv.org/pdf/1508.00335)  for a clear discussion.

(b) **Technical concerns**

- Discussion/Comparison with related work

Authors in related work, such as Paranjape et al 2020, Guerreiro et al 2021, highlighted several points in the limitation of sample-wise feature selection. I believe that concrete empirical comparisons are required to clearly demonstrate the benefits of the proposed approach. It is important to note that the purpose of this is to consider additional empirical evidence to support your claim. And there's no need to bring up the recent SOTA.

---

> ### Author Response · Authors · 2023-03-03
> **We appreciate your review (part 1)**
>
> Thank you for your interesting comments. We have updated our work according to the responses below:
>
> > Overall structure.
>
> Thanks for the suggestion. Following your suggestion, we have updated the methods section to include an overview of the proposed method (see the beginning of Sec. 3), and then provide further details for important constituents of Algorithm 1. We hope that this will help better relate the overall algorithm with the corresponding details in Sec. 3. We have also annotated steps in Algorithm 1 with the section numbers that contain the corresponding details, which facilitates cross-referencing. Per your feedback, we have also reduced the length of Sec. 3.
>
> > math notation.
>
> We define Sparsemax in Eq. 1, and the sparse feature mask $\mathbf{m_{sp}}$, defined in Eq. 2, is the result of sparsemax applied to the non-sparse mask $\mathbf{m}$. We have moved the hint that $\mathbf{m_{sp}}$ is a vector that lives in $\mathbb{R}^{d}_{\ge0}$ to the subsequent paragraph to simplify the expression in Eq. 2. We have also updated both Figure 1 and its caption to include more information and clarification. Please let us know if these do not sufficiently address your concerns on clarity.
>
> Regarding the notation in Lemma 3.2 and in Eq. 4, all notations that are not specified in the notations section are specific to Lemma 3.2 and in Eq. 4 and defined there. We specifically did not make the notations section very long to improve readability. Among these notations specific to Lemma 3.2 and Eq. 4, $\mathbb{v}$ is a $d$-dimensional non-uniform vector, $F$ indicates the number of nonzero elements in sparsemax output, and $v_{(i)}$ is an element of $\mathbb{v}$, where $i$ indicates the index of $v_{(i)}$ amongst the sorted values of $\mathbb{v}$. In Eq. 4, $F_t$ denotes the number of selected features at step $t$, and $N$ the total number of training steps as denoted in the notations section.
>
> > mutual information.
>
> We have clarified this point, thank you for pointing it out. There is a long line of research on mutual information and entropy estimation that focuses on the case where the random variables live in the discrete space [Paninski 2003, Kraskov et al 2004, Valiant and Valiant 2011, Han et al 2015, Jiao et al 2015, Wu and Yang 2016]. This is because 1) many variables in machine learning are indeed discrete, e.g. vocabulary indices for text data used in NLP, categorical variables such as whether transaction is fraudulent, etc, and 2) MI estimation in the continuous case can be reduced to the discrete case via binning and taking a limit [Kraskov et al 2004, Paninski 2003], as done in Sec. A.4.
>
> References:
>
> Alexander Kraskov, Harald Stögbauer, and Peter Grassberger, Estimating mutual information. PHYSICAL REVIEW E 69, 066138 (2004).
>
> Liam Paninski. Estimation of Entropy and Mutual Information. Neural Computation, 15, 1191–1253 (2003).
>
> Yanjun Han, Jiantao Jiao, and Tsachy Weissman. Adaptive estimation of Shannon entropy. In Information Theory (ISIT), pages 1372–1376. IEEE, 2015.
>
> Jiantao Jiao, Kartik Venkat, Yanjun Han, and Tsachy Weissman. Minimax estimation of functionals of discrete distributions. IEEE Transactions on Information Theory, 61(5):2835– 2885, 2015.
>
> Gregory Valiant and Paul Valiant. Estimating the unseen: an $n/\log (n)$-sample estimator for entropy and support size, shown optimal via new CLTs. In Proceedings of the forty-third annual ACM symposium on Theory of computing, pages 685–694. ACM, 2011.
>
> Yihong Wu and Pengkun Yang. Minimax rates of entropy estimation on large alphabets via best polynomial approximation. IEEE Transactions on Information Theory, 62(6):3702–3720, 2016.
>
> > Objective $E(X, Y)$.
>
> This is a quadratic error function, it gives the expected square error of the model predictions $R(x, y)$, as defined in equation set 7. It is not intended to be an energy term (the letter E might cause this confusion, but it comes from the word “error”). This is a generalization of the commonly used mean squared error function, where instead of averaging, we take the expectation of the square error $(1-R(x,y))^2 + \sum_{y’\in \mathcal{Y}\y} R(x, y’)^2$ with the joint probability $P_{X, Y}(x, y)$ for each $(x, y)$ pair.
>
> $y$ is indeed continuous in the regression setting, we briefly addressed this in Sec. 4.2 on page 6, and have introduced Sec. A.4 to provide detailed derivations in this case.
>
> (continued below)

---

> > ### Author Response · Authors · 2023-03-03
> > **We appreciate your review (part 2)**
> >
> > > sparsemax vs softmax.
> >
> > Softmax would indeed be another way to output probabilities, which we consider in Sec. A.5. Regarding your points: note that the scaling parameter used to scale sparsemax input (Lemma 3.2) has a different nature from the typical use of temperature in softmax. The scaling term controls exactly the number of nonzero elements in sparsemax output, i.e. the output will already be _on the probability simplex with the desired number of nonzero elements_, without an additional normalization step. However, one needs to threshold the softmax output regardless of whether a temperature term is added; the temperature term would only change the relative distance between output elements, not their ordering. This threshold would need to be determined individually post hoc after each softmax evaluation for each sample. Furthermore, there does not exist a temperature such that the top-k elements in softmax output will already be on the probability simplex (for $k<$ output dimension), since the entire output normalizes to 1, hence an additional normalization step is required, as opposed to sparsemax.
> >
> > Sparsemax is dominated by sorting, and hence has complexity $O(F_0 \log F_0)$ per sample, where $F_0$ is the total number of features (further details in section 4.4). For $n$ samples, this gives a total complexity of $O(nF_0 \log F_0)$.
> >
> > In addition to these, as noted in section A.5, sparsemax also has the advantage of an additional inductive bias for feature selection: $sparsemax(v)$ is more equi-distributed over the interval $[0,1]$ than softmax-top-k normalization (i.e. $sparsemax(v)$ has lower entropy than softmax-top-k normalization), making it more discriminatory for feature selection.
> >
> > > gradient computation
> >
> > Imposing an $\ell_1$ norm constraint (a commonly-used idea to achieve sparsity for model weights) would not produce a projection onto the probability simplex of the target dimension, which sparsemax directly achieves. Sparsemax projection is discussed in Sec. 3.1. (Eq. 1 describes the sparsemax optimization objective. Eq. 4 refers to the schedule for tempering the number of features.)
> >
> > > Approximation of KL divergence vs f divergence.
> >
> > Our claim that $I_q(X, Y)$ in Eq. 6 is an approximation of the mutual information formulation defined in Eq. 5 is justified in Sec. 4.1. We have reviewed the reference you provided. Our claim does not contradict your claim of Eq. 6 being an approximation of Eq. 5 defined via KL divergence, since KL divergence is a type of f divergence, where the $f$ function is the logarithm (please see https://people.lids.mit.edu/yp/homepage/data/LN_fdiv_short.pdf for further details).
> >
> >
> > > Technical concern: discussion with related work, e.g. Paranjape et al 2020, Guerreiro et al 2021.
> >
> > Indeed we cite both Paranjape et al 2020 and Guerreiro et al 2021 for their work on sample-wise masking, which they use for rationale extraction. As you said, there are many limitations to sample-wise feature selection. Our work focuses on global feature selection, with more general applicability than rationale extraction. Global feature selection can judiciously reduce the model capacity by selecting the most salient features consistent across all the samples. Therefore, our numerous experiments in both the experiments section and the appendix cover 10 relevant baselines for global feature selection, showing strong results for SLM.
> >
> > We believe these responses and the updated paper addressed the points you raised. We are happy to hear additional suggestions or comments.

---

> > ### Comment · Reviewer_24kh · 2023-03-25
> > **Follow-up**
> >
> > Dear authors,
> >
> > I am just wondering that you finished the rebuttal? I saw
> > > (continued below)
> >
> > but there is no following-up. If it is done, I will check them in the following days..
> >
> > Thanks

---

> > > ### Author Response · Authors · 2023-03-25
> > > **Thank you for noticing this, corrected visibility.**
> > >
> > > Dear Reviewer 24kh,
> > >
> > > Thank you very much for checking in, indeed we posted part 2 of the response within a minute of part 1 on March 3, but for some reason the readers were set to "Editors In Chief, Action Editors, Authors" rather than "Everyone". We just updated the audience to "Everyone", and it should be visible now.
> > >
> > > Thanks again for letting us know, please let us know if you have further comments or suggestions.

---

> ### Comment · Reviewer_24kh · 2023-04-04
> **Update after checking rebuttal**
>
> Dear authors,
>
> Thanks for your hard work. After checking rebuttal, revised paper and others' reviews. I would think this paper may still require major revisions. My further thoughts are as follows.
>
> - (Important) The main objective of this paper is to provide an explainable machine learning model through differentiable feature selection. However, except numeric values, there is no clear evidence (such as concrete samples) to illustrate how the feature selection looks like (this is also raised by other reviewer). I do think this is crucial in explainable machine learning, numeric values are indeed insufficient to make the method explainable.
>
> - This paper used sparsemax, which requires projection operation on a high-dimensional dataset, while I still think this operation could be replaced by simple softmax+some temperature scaling. The rebuttal proposed by the author is still not sufficiently convincing.
>
> - In the proposed loss function, this paper supposed it could work in both classification and regression. This reviewer still feels quite confusing in the regression setting since the operation $\sum_{y} f(y)$ over the continuous variable $y$ is not validated (it should be an integral).
>
> - Besides, I agree some technical concerns raised by Reviewer kDMo. Hopefully it could be addressed.
>
> Overall, I appreciate the simplicity of the proposed approach. I still encourage authors conduct a major revision and resubmit.

---

> > ### Author Response · Authors · 2023-04-05
> > **Some clarification points.**
> >
> > We would like to add some clarification points:
> >
> > 1. SLM learns a sparse mask used for feature selection, as described in Algorithm 1. The features that have the mask coefficient of 0 are not selected, i.e. they are not used by the model. The methods section (Section 3) describes the form and use of the learned sparse mask vector $\mathbf{m_{sp}} \in \mathbb{R}^d$, which zeros out non-selected features via an element-wise product: $\mathbf{X} \odot \mathbf{m_{sp}}$. The numeric values presented gave examples of the learned mask and the feature selection process. We will include a figure dedicated to this in an updated version.
> >
> > 2. Can you describe what aspect of the use of sparsemax is not convincing? In addition to the response above, softmax is inherently not sparse, even with a small temperature, i.e. when the probability mass concentrates on the top few classes, the probability masses away from the top classes are still non-zero, contrary to what sparsemax achieves. Furthermore, there does not exist a temperature such that the top-k elements in softmax output will already be on the probability simplex, (since the entire softmax output normalizes to 1), again contrary to what sparsemax achieves.
> >
> > 3. We discuss the continuous vs discrete cases at the beginning of Section 4, at the end of Section 4.3, and in Section A.4. We do not suppose that the same loss function works for both continuous and discrete cases. Indeed the summation should become an integral in the continuous case, this is done in Section A.4. We build on an established line of literature that focuses on the discrete variables case, as discussed in Section 4.
> >
> > The technical concerns raised by Reviewer kDMo have been addressed. We would be grateful if you could review that.

---

### Review · Reviewer_byyD · 2023-03-07

**Summary Of Contributions:**

This paper studies the feature selection problem. An MI-based method is proposed to select features. The experiments are conducted on many datasets and verify the effectiveness of the proposed method. Specifically, the proposed method learns which features to select, and gives rise to a novel objective that provably maximizes the mutual information (MI) between the selected features and the labels, which can be derived from a quadratic relaxation of mutual information from first principles. In addition, the authors derive a scaling mechanism that allows SLM to precisely control the number of features selected, through a novel use of sparsemax.

**Audience:**

Yes

**Broader Impact Concerns:**

There is not concern regarding broader impacts.

**Claims And Evidence:**

No

**Requested Changes:**

Mutual information is just one measure to see the independency between two samples. It would be better to include more measures to verify the effectiveness of the proposed method. For example, HSIC can be directly used for the purpose and can form a good baseline for this paper.

**Strengths And Weaknesses:**

Pros:

This paper is easy to follow, and the proposed method makes sense to select impactful features.

This paper considers using many datasets to verify their method's performance.

Cons:

1. Mutual information is just one measure to see the independency between two samples. It would be better to include more measures to verify the effectiveness of the proposed method. For example, HSIC can be directly used for the purpose and can form a good baseline for this paper.

In general, this paper is solid. However, adding maximizing HSIC as a baseline would be better to improve the impact of this paper.

---

> ### Author Response · Authors · 2023-03-14
> **Thank you for your suggestions.**
>
> Thank you for the suggestions in your review.
>
> We note that HSIC-Lasso is indeed an existing baseline, which integrates the HSIC objective into nonlinear feature selection. SLM was shown to outperform HSIC-Lasso all datasets (please see Table 1).
>
> In addition, following your suggestion, we have introduced a new version of SLM with the HSIC objective to demonstrate the effectiveness and generalizability of SLM’s learned feature selection mask. Essentially, we consider replacing the MI objective with the HSIC objective, as another measure of dependency between the input features random variable and the labels random variable. We have implemented HSIC following the original formulation by Gretton et al. 2005 and added a new section Appendix A.8. We condition the HSIC objective on the learned feature selection mask by scaling the standard-normalized input features by the feature selection mask, before sparsemax normalization is applied. We have experimented to find the best kernel function for the HSIC objective.
>
> For the comparisons, we have performed hyperparameter tuning similar to the ones in our main experiments, and found that SLM learned feature mask combined with the HSIC objective perform very competitively compared to the myriad of baselines in the main experiments, demonstrating the effectiveness and generalizability of SLM’s learn feature selection mask approach.
>
> Note that unlike mutual information, HSIC is not a probabilistic measure, as such it cannot organically take advantage of the interpretation that the learned feature selection mask is a probability vector (which indicates the probability that a feature is selected, and is also useful for feature importance interpretability), the way that the MI objective does in Eq. 8.
>
> We would be happy to address any additional comments or questions.

---

### Decision · Action_Editors · 2023-04-03

**Recommendation:** Reject

**Comment:**

The submission proposed sparse learnable masks for end-to-end feature selection, which tries to maximize the mutual information between the input and output random variables. While there are many interesting points in the submission, two reviewers found that there are still some conceptual and technical issues after the rebuttal and then they voted for rejection in our internal discussions (see https://openreview.net/forum?id=qcXwX7CJvX&noteId=MBloPbgAXE and https://openreview.net/forum?id=qcXwX7CJvX&noteId=FEiWYKzKTx for details, which I asked the reviewers to post and which are almost identical to their internal discussions).

In particular, I agree with them in the evaluation of the proposal. The goals of feature selection and feature extraction (nowadays representation learning) are different, where feature selection is to better understand the data and to provide an explainable model, and feature extraction is to outperform states of the art on certain tasks. The reviewers argued that except numeric values, there is no clear evidence to illustrate how the feature selection looks like, and numeric values are indeed insufficient to make the method explainable.

Therefore, we cannot accept the submission for publication in its current version, and please take the comments in your revision.

**Audience:**

Yes, feature selection is a practically very important problem.

**Claims And Evidence:**

Partially, there are still some conceptual and technical issues after the rebuttal.